# UNIEDIT: A Unified Knowledge Editing Benchmark for Large Language Models

**Qizhou Chen**[1,3], **Dakan Wang**[2], **Taolin Zhang**[4], **Zaoming Yan**[1], **Chengsong You**[1], **Chengyu Wang**[3*], **Xiaofeng He**[1*]

[1]East China Normal University, Shanghai, China [2]Exacity Inc., Shanghai, China
[3]Alibaba Group, Hangzhou, China [4]Hefei University of Technology, Hefei, China
52265901009@stu.ecnu.edu.cn

## Abstract

Model editing aims to efficiently revise incorrect or outdated knowledge within LLMs without incurring the high cost of full retraining and risking catastrophic forgetting. Currently, most LLM editing datasets are confined to narrow knowledge domains and cover a limited range of editing evaluation. They often overlook the broad scope of editing demands and the diversity of ripple effects resulting from edits. In this context, we introduce UNIEDIT, a unified benchmark for LLM editing grounded in open-domain knowledge. First, we construct editing samples by selecting entities from 25 common domains across five major categories, utilizing the extensive triple knowledge available in open-domain knowledge graphs to ensure comprehensive coverage of the knowledge domains. To address the issues of generality and locality in editing, we design an Neighborhood Multi-hop Chain Sampling (NMCS) algorithm to sample subgraphs based on a given knowledge piece to entail comprehensive ripple effects to evaluate. Finally, we employ proprietary LLMs to convert the sampled knowledge subgraphs into natural language text, guaranteeing grammatical accuracy and syntactical diversity. Extensive statistical analysis confirms the scale, comprehensiveness, and diversity of our UNIEDIT benchmark. We conduct comprehensive experiments across multiple LLMs and editors, analyzing their performance to highlight strengths and weaknesses in editing across open knowledge domains and various evaluation criteria, thereby offering valuable insights for future research endeavors. [1]

## 1 Introduction

Large language models (LLMs), with their powerful natural language processing capabilities, have sparked a revolution in the field of artificial intelligence and have become indispensable tools across various industries, including medicine [1–3], finance [4–6], and education [7–9]. However, as application scenarios continue to expand or as these models function in ever-changing environments, they often fail to provide sufficiently accurate and real-time information [10–12]. This can have significant impacts on high-risk and high-demand industries. Model editing techniques aim to efficiently and precisely update the internal knowledge of these models while avoiding the computational burden and catastrophic forgetting typically associated with retraining LLMs [13, 14].

The interpretability of knowledge localization in transformers [15, 16] has provided practical motivation for model editing. Consequently, model editing can be achieved by modifying these parameters [16–19]. In these early works, benchmarks like ZSRE [20] and CounterFact [16] evaluate whether

---

*Co-Corresponding Authors

[1]Code and dataset are available at `https://github.com/qizhou000/UniEdit` and `https://huggingface.co/datasets/qizhou/UniEdit`.

39th Conference on Neural Information Processing Systems (NeurIPS 2025) Track on Datasets and Benchmarks.

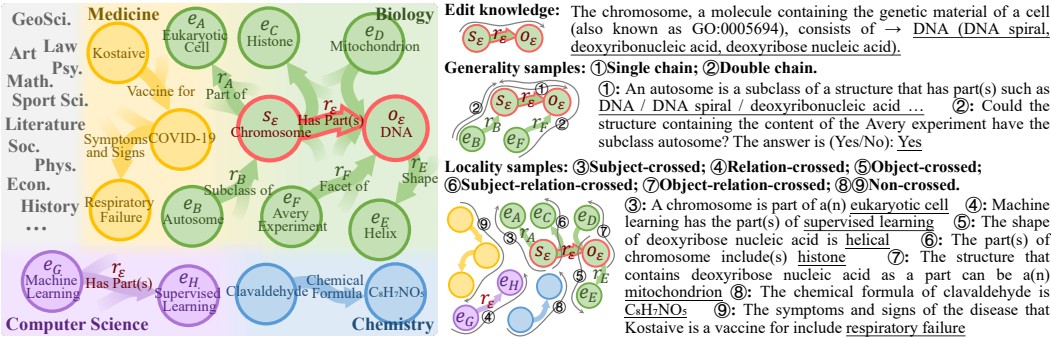

Figure 1: Data composition of UNIEDIT, covering up to 25 different domains extracted in Wikidata. Given an editing triple (highlighted with a red edge), generality and locality structures are sampled as multi-hop chain subgraphs from its neighborhood. The generality subgraphs include the entire editing triple, while locality refers to all other cases. In each subgraph, a node is selected to serve as the cloze target, forming a single chain prompt if it is an endpoint, or a double chain prompt otherwise (only single chain shown for locality here). Beyond prompt structural differences, locality samples further classified into six types according to their cross-features with the editing triple (see Appendix A.2 for correspondence with the criteria).

LLMs after editing incorporated the edited content (i.e., reliability) and its paraphrased versions (i.e., generality), while preserving the responses to all other inputs as before editing (locality). These two datasets focus solely on the edit samples themselves, but did not evaluate the queries which are indirectly related to the edited facts [21]. Moreover, editors are prone to giving LLMs high confidence in the edited content [22]. This will lead to errors on samples which have some overlap with the edited content but are actually irrelevant. As a result, various editing datasets have been proposed to evaluate various generality types, such as multi-hop [21], subject aliasing [23], and relation reversal [24]; and locality types, such as subject specificity [22], relation specificity [22], and 1-N forgetfulness [23]. However, most existing benchmarks only evaluated generality and locality on a limited set of knowledge domains. They are typically sampled from a few triples in Knowledge Graphs (KGs) and limited to a small number of relations [21, 23–26], or refined from other editing datasets [22, 27, 28]. The evaluation result on such a restricted dataset may not imply the same conclusion for a diverse set of knowledge domains. Furthermore, each benchmark independently constructs data based on its proposed evaluation criteria, lacking a dataset to integrate all of them. Such a dataset would enable the evaluation of various combinatorial cases and entail potential new challenges. For example, a generality sample may simultaneously encompass multiple hops [21], relation reversal [24], and subject alias [23].

Based on the above analysis, we introduce a new LLM knowledge editing benchmark, UNIEDIT, designed to evaluate and enhance the editing capabilities and generalization robustness of LLM editors in open-domain knowledge. We use Wikidata [29], the largest open-source KG, to build this benchmark, as shown in Figure 1, which illustrates its composition and example cases. After data cleaning, we obtain 29.9M entities and 2,500 properties (relations in the KG) from a total of 113.7M entities and 12,300 properties. We categorize entities by domains to ensure balanced coverage across various areas. Subsequently, these entities are used to sample knowledge triples and subgraphs for data generation.

To address the limitation of existing benchmarks with single evaluation criteria, we propose the Neighborhood Multi-hop Chain Sampling (NMCS) algorithm to construct more diverse and comprehensive edit evaluations for generality and locality. Each dataset sample is generated by converting a knowledge subgraph into natural language, enabling structurally controlled semantics. Given a triple selected for editing data generation, NMCS ensure that the subgraph of a generality sample contains the entire editing triple. In contrast, subgraphs that exclude or partially contain components of the editing triple are considered locality samples. Building on this unified sampling scheme, our NMCS algorithm integrates and extends evaluation criteria from previous benchmarks, making the dataset significantly more challenging.

UNIEDIT is the first open-domain knowledge editing benchmark designed to comprehensively simulate a wide range of knowledge editing challenges encountered in the real world. Table 1 illustrates the features encompassed by existing benchmarks. Our main contributions are as follows.

Table 1: Coverage of various features in existing benchmarks, including Rephrase (Rep), Multi-Hop (MH), Relation Reversal (RR), Same Entity Reasoning (SER), Subject Alias (SA), Object Alias (OA), Subject Specificity (SS), Relation Specificity (RS), Object Specificity (OS), 1-N Forgotten (1-NF), Combinations of the above evaluation Criteria (CC), and Open-Domain (OD). See Appendix A for detailed definitions and instances of these evaluation criteria.

| Benchmarks | Generality | | | | | | Locality | | | | CC | OD |
|---|---|---|---|---|---|---|---|---|---|---|---|---|
| | Rep | MH | RR | SER | SA | OA | SS | RS | OS | 1-NF | | |
| ZSRE [20] | ✓ | | | | | | | | | | | |
| CounterFact [16] | ✓ | | | | | | | ✓ | | | | |
| MQuAKE [21] | ✓ | ✓ | | | | | | | | | | |
| BAKE [24] | ✓ | | ✓ | | | | | ✓ | | | | |
| RippleEdit [23] | | ✓ | ✓ | | | ✓ | | ✓ | | ✓ | | |
| ReCoE [25] | | | | ✓ | | | | | | | | |
| EVOKE [22] | ✓ | ✓ | | | | | ✓ | ✓ | | | | |
| CliKT [30] | ✓ | | | | | | | | | ✓ | | |
| HalluEditBench [31] | ✓ | ✓ | | | | | ✓ | | | | | |
| WikiBigEdit [32] | ✓ | ✓ | | | | | | ✓ | | | | |
| UNIEDIT | ✓ | ✓ | ✓ | ✓ | ✓ | ✓ | ✓ | ✓ | ✓ | ✓ | ✓ | ✓ |

- We provide a complete pipeline and toolkit for converting Wikidata, the largest open-source and continuously updated open-domain KG, into a knowledge editing dataset.

- To tackle the diversity of ripple effects entailed by edits, we introduce the NMCS algorithm to unify and extend various evaluation criteria, thereby achieving more diverse and general editing evaluation coverage.

- We conduct extensive editing experiments on multiple LLMs, applying various editing methods within UNIEDIT. The experimental results and corresponding analyses offer valuable insights into the performance and limitations of existing LLM editors.

## 2 Related Works

In this section, we briefly summarize the related works on knowledge editing.

### 2.1 Knowledge Editing Methods

Knowledge editing methods can be categorized into two main approaches: Locate-then-Edit (L&E) methods and external module-based strategies.

**L&E:** ROME [16] identifies edit-sensitive layers via causal tracing for targeted layer weight updates. MEMIT [17] and WILKE [18] enhance ROME by distributing changes across multiple layers. PMET [19] employs information extraction patterns of attention layers to implement precise updates. AlphaEdit [33] extends L&E to lifelong editing through zero-space projection strategies. UnKE [28] and AnyEdit [34] explore strategies for adapting L&E methods to unstructured knowledge editing.

**External Modules:** KE [35] trains an LSTM-based hyper-network to predict parameter updates given edit samples. Compared to KE, MEND [36] enhances the editing signal by using the first-order gradient of the edit knowledge. SERAC [37] trains a counterfactual model for query redirection. T-Patcher [38] incorporates additional neurons for edited knowledge, and GRACE [39] remaps edit-related representations based on edit distance thresholds. RECIPE [40] creates continuous prefixes for dynamic editing through prompt learning. LEMOE [41] enables lifelong editing using a Mixture of Experts with expert routing.

Beyond the two mainstream paradigms, early efforts such as ENN [42] investigated model editing through meta-learning. [43] explicitly introduced knowledge editing for large transformers and explored partial parameter tuning to achieve it. [15] proposed the concept of "knowledge neurons" and studied how factual knowledge is stored in pretrained transformers, providing practical motivation for the L&E paradigm. Furthermore, IKE [44] uses in-context learning to enable LLMs to follow editing instructions.

## 2.2 Knowledge Editing Benchmarks

In earlier work, ZSRE [20] utilizes WikiReading [45] to generate QA editing data, while Counterfact [16] constructs counterfactual data to increase difficulty. These efforts focus on evaluating whether LLMs recall the edited knowledge and its paraphrased versions but overlook the ripple effects induced by the edits. To address this, MQuAKE [21] and BAKE [24] explore multi-hop and relational reversal questions. RippleEdit [23] refines the definition of multi-hop, further identifying 1-N forgetfulness, entity aliasing, and relation specificity. ReCoE [25] investigates entity reasoning to examine LLMs' ability to apply edited knowledge. EVOKE [22] assesses the overfitting problem of L&E methods by reassessing existing benchmarks. CliKT [30], HalluEditBench [30], and WikiBigEdit [32] construct editing datasets focused on biomedical long-tail knowledge, hallucinations within LLMs, and recently updated Wikidata knowledge, respectively. Additionally, AKEW [27], UnKEBench [28], and AnyEdit [34] address unstructured editing in texts without subject localization for L&E methods. While relevant, our work primarily focuses on edited knowledge domains and their induced effects. Unstructured cases can be implemented simply by omitting subject positions or merging texts.

Despite these efforts, existing benchmarks remain limited by their narrow knowledge domains, one-sided evaluation criteria, and generally small scale. These limitations will restrict the future development of editors, especially for methods that require edit training [36, 37, 40]. UNIEDIT effectively addresses these challenges.

## 3 Problem Definition

Factual knowledge can be represented as a triple $(s, r, o)$, consisting of a head entity (subject) $s$, a relation $r$, and a tail entity (object) $o$. Given an LLM $f_{llm} \in \mathcal{F}$ as a function $f_{llm} : \mathcal{Q} \mapsto \mathcal{O}$ that maps an input query $q$ to its output $o = f_{llm}(q)$, an editing request $\varepsilon = (q_\varepsilon, o_\varepsilon)$ instructs the LLM to recall $o_\varepsilon$ based on $s_\varepsilon$ and $r_\varepsilon$. Here, $q_\varepsilon = f_{nl}(s_\varepsilon, r_\varepsilon)$ is the natural language prefix derived from the editing subject $s_\varepsilon$ and relation $r_\varepsilon$, while $o_\varepsilon$ is the object to be edited.

Given a collection of edits $\mathcal{E} = \{\varepsilon_i\}_{i=1}^{\tau}$, model editing involves designing an editor $\mathbf{ME} : \mathcal{F} \times \mathcal{Q} \times \mathcal{O} \mapsto \mathcal{F}$ that produces a post-edit LLM $f'_{llm} = \mathbf{ME}(f_{llm}, \mathcal{E})$. The edited $f'_{llm}$ should meet the following three metrics [10] (see Appendix A for more details):

**Reliability** requires that $f'_{llm}$ recalls the edited knowledge itself, i.e., $f'_{llm}(q_{\varepsilon_i}) = o_{\varepsilon_i}$ for $i = 1, \ldots, \tau$.

**Generality** requires that $f'_{llm}$ adjusts responses for queries related to edited samples, i.e., $f'_{llm}(q_g) = o_g$, where $(q_g, o_g) \sim \mathcal{G}(\mathcal{E})$. Here, $\mathcal{G}(\mathcal{E})$ is the relevant neighborhood of the edit collection $\mathcal{E}$.

**Locality** requires $f'_{llm}$ to maintain consistency with the initial model $f_{llm}$ on queries unrelated to previously edited knowledge, i.e., $f'_{llm}(q_l) = f_{llm}(q_l)$, where $(q_l, o_l) \sim \mathcal{L}(\mathcal{E})$. Here, $\mathcal{L}(\mathcal{E})$ represents the sample distribution independent of $\mathcal{E}$, excluding $\mathcal{E} \cup \mathcal{G}(\mathcal{E})$.

A good editor should allow the LLM to adapt to different levels of generality and locality based on the semantics of the edit.

## 4 UNIEDIT

This section introduces the data construction and statistics of UNIEDIT.

### 4.1 Data Construction Process

The overall process of data construction, as shown in Figure 2, is introduced below. For more details, please refer to Appendix B.

**Step 1. Data Preparation and Cleaning:** The Wikidata full export `latest-all.json`[2] contains 113.7M entities and 12,300 properties (relations), each of which has an ID, label, data type (only for properties), description, aliases, and claims. The data type indicates the type of value that the property points to. The claims indicate all triples in which the entity acts as the head, listing properties and their corresponding values (tails). We filter out entities with no English labels and remove those containing low-utility keywords in their descriptions (like "point of time"), reducing the total to 29.9M items.

---

[2]Download from `https://dumps.wikimedia.org/wikidatawiki/entities/`

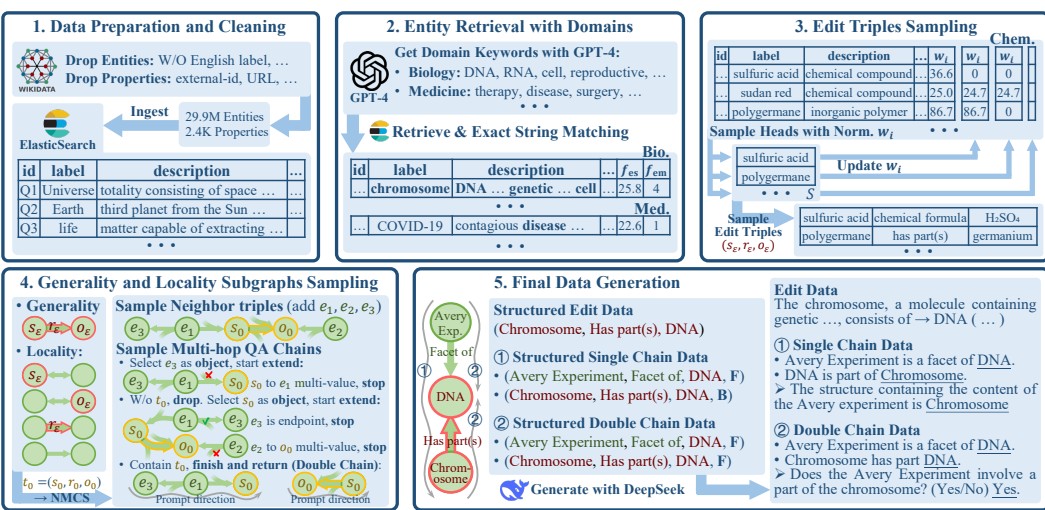

Figure 2: Data construction pipeline of UNIEDIT. Steps 1–3 include data preprocessing, domain-specific entity retrieval, and sampling of relevant triples based on the domain entity. In Step 4, generality and locality QA chains are sampled using NMCS algorithm. In Step 5, the final data is generated based on the sampled QA chains, where **F** and **B** indicate the forward and backward directions, respectively—referring to the prompt generation direction with respect to the triple.

Then, we ingest the cleaned entities into search engine (Elasticsearch [46]) for subsequent retrieval and sampling. For properties, through data type filtering and manual verification, 2.4K items are retained after removing nonlinguistic and low-editorial-value items (e.g., those pointing to images, IDs, and URLs). The retained properties fall into seven types: wikibase-item (pointing to entities), string, quantity, time, math, globe-coordinate, and monolingual text.

**Step 2. Entity Retrieval with Domains:** We categorize entities by domain to ensure balanced coverage across fields, promoting the openness and diversity of knowledge sampling. Five sectors are considered: Natural Sciences, Humanities, Social Sciences, Applied Sciences, and Interdisciplinary Studies, collectively covering up to 25 domains (Figure 3a). Using domain-specific keywords generated by GPT-4 [47], we retrieve relevant entities based on their labels and descriptions. To improve relevance, we further apply exact string matching to filter out noisy results caused by the tokenization of search engine (e.g., "black hole" matching only "black").

**Step 3. Edit Triples Sampling:** A domain entity set $E = \{e_i\}$ is quite large, making it impractical to construct edit triples for all of them. Therefore, we first sample head entities from $E$ for edit triples. To ensure diversity, we use a sequential weighted sampling approach and dynamically adjust the sampling weight to reduce the likelihood of semantically similar items to be sampled in later stage. Let $\sigma(X, P)$ denote a sampling function that returns an element $x_i \in X$ with probability $p_{x_i} \in P$. Initializing the head entity set as $S = \emptyset$, the sampling process proceeds step by step as $S = S \cup \{\sigma(E, P_E)\}$, where each probability $p_{e_i}$ in the distribution $P_E = \{p_{e_i}\}$ is given by:

$$p_{e_i} = \frac{w_i}{\sum_j w_j} \text{ s.t. } w_i = \begin{cases} 0, & \text{if } e_i \in S, \\ f_{\text{iw}}(e_i)/\gamma^{\psi(e_i,S)}, & \text{else.} \end{cases} \tag{1}$$

where we define the initial sampling weight of an entity, based on its domain relevance, as $f_{\text{iw}}(e_i) = f_{\text{es}}(e_i)f_{\text{em}}(e_i)$. Here, $f_{\text{es}}(e_i)$ is the ElasticSearch retrieval score, and $f_{\text{em}}(e_i)$ is the exact match count of domain keywords in the description of $e_i$. This combination heuristic balances between partial and exact matches. $\gamma$ is the decay base, set as 1.05. $\psi(e_i, S) = \sum_{s \in S} \text{sim}(e_i, s)$ is the decay factor to down-weight the sampling probability of $e_i$ based on its average similarity with the already sampled items. Formally,

$$\text{sim}(e_i, s) = \sum_{u_{e_i} \in f_{\text{dw}}(e_i)} \sum_{u_s \in f_{\text{dw}}(s)} \frac{\mathbb{I}(u_{e_i} = u_s)}{\|f_{\text{dw}}(e_i)\|} \delta(u_s) \text{ s.t. } \delta(u) = \begin{cases} \delta_{\text{in}}, & \text{if } u \in U, \\ \delta_{\text{out}}, & \text{else.} \end{cases} \tag{2}$$

where $\mathbb{I}$ is the indicator function. $f_{\text{dw}}(e)$ denotes the set of word segments extracted from the description of entity $e$, and $\delta(u)$ is the decay weight based on the domain keyword set $U$. To mitigate

the impact of sampling decay on domain relevance, we assign a lower decay weight $\delta_{\text{in}}$ to words in $U$. Specifically, we set $\delta_{\text{in}} = 0.2$ and $\delta_{\text{out}} = 1$. Intuitively, the more word segments in the description of $e_i$ are covered by those in $S$, the lower its sampling priority should be.

A total of 30,000 head entities are sampled for each domain in our work. Given a sampled head entity $s_\varepsilon$, the edit triple $t_\varepsilon = \sigma\left(f_{\text{twh}}(s_\varepsilon), \mathcal{U}\right)$ is generated, where $f_{\text{twh}}(s_\varepsilon)$ represents all the triples with $s_\varepsilon$ as the head, which are obtained by traversing the properties and corresponding values in the claims field of $s_\varepsilon$. $\mathcal{U}$ represents uniform distribution. Since the properties filtered out during the initial cleaning step are omitted, the function may return an empty set.

---

**Algorithm 1** Neighborhood Multi-hop Chain Sampling (NMCS)

---

**Input:** Initial triple $t_0 = (s_0, r_0, o_0)$, triples to be excluded $\bar{T} = \{\bar{t}_i\}$, maximum sampling attempts $m$, maximum hops $h$, the entity full set $\tilde{E}$.
**Output:** Multi-hop chains $\mathcal{T}$

1: $T = \{t_0\}$ *# Subgraph triple set*
2: $E_{\text{add}} = \{s_0\}$ *# Added nodes*
3: **if** $o_0 \in \tilde{E}$ **then** $E_{\text{add}} = E_{\text{add}} \cup \{o_0\}$
4: $E_{\text{end}} = \text{clone}(E_{\text{add}})$ *# End nodes*
5: *# Expand both sides of $t_0$ to sample a chain of neighboring triples*
6: **while** $\text{len}(T) < h$ **and** $\text{len}(E_{\text{end}}) > 0$ **do**
7: $\quad e = \sigma\left(E_{\text{end}}, \mathcal{U}\right)$
8: $\quad$ **for** $i = 1$ **to** $m$ **do**
9: $\qquad f_{\text{tw}^*} = \sigma\left(\{f_{\text{twh}}, f_{\text{twt}}\}, \mathcal{U}\right)$
10: $\qquad t = \sigma(f_{\text{tw}^*}(e), \mathcal{U})$
11: $\qquad$ **if** $t = \varnothing$ **or** $t \in \bar{T}$ **then continue**
12: $\qquad (s, r, o) = t$
13: $\qquad$ **if** $\{s, o\} \cap E_{\text{add}} = \{e\}$ **then**
14: $\qquad\quad$ **break** *# Acyclic, finish sampling*
15: $\quad E_{\text{end}} = E_{\text{end}} \setminus \{e\}$
16: $\quad$ **if** $t = \varnothing$ **then continue**
17: $\quad T = T \cup \{t\}$
18: $\quad$ *# Update added nodes and end nodes*
19: $\quad$ **if** $f_{\text{tw}^*} = f_{\text{twt}}$ **then**
20: $\qquad E_{\text{add}}, E_{\text{end}} = E_{\text{add}} \cup \{s\}, E_{\text{end}} \cup \{s\}$
21: $\quad$ **else**
22: $\qquad E_{\text{add}} = E_{\text{add}} \cup \{o\}$
23: $\qquad$ **if** $o \in \tilde{E}$ **then** $E_{\text{end}} = E_{\text{end}} \cup \{o\}$
24: *# Map entities to triples*
25: $M = \text{defaultdict}(\text{list})$
26: **for** $t$ **in** $T$ **do**
27: $\quad (s, r, o) = t$
28: $\quad M[s].\text{append}(t), M[o].\text{append}(t)$
29: *# Randomly select object $e$ and expand both sides to construct valid multi-hop QA chains*
30: **for** $e$ **in** $\text{shuffle}(\text{list}(E_{\text{add}}))$ **do**
31: $\quad \mathcal{T} = [[t]$ **for** $t$ **in** $M[e]]$
32: $\quad$ **for** $C$ **in** $\mathcal{T}$ **do**
33: $\qquad e_{\text{ce}} = e$ *# Current end to extend chain*
34: $\qquad$ **while true do**
35: $\qquad\quad (s, r, o) = C[-1]$
36: $\qquad\quad e_{\text{ce}} = s$ **if** $s \neq e_{\text{ce}}$ **else** $o$
37: $\qquad\quad$ **if** $\text{len}(M[e_{\text{ce}}]) = 1$ **then**
38: $\qquad\qquad$ **break** *# Endpoint*
39: $\qquad\quad t_1, t_2 = M[e_{\text{ce}}]$
40: $\qquad\quad (s, r, o) = t = t_1$ **if** $t_1 \neq C[-1]$ **else** $t_2$
41: $\qquad\quad$ **if** $e_{\text{ce}} = s$ **and** $\|f_{\text{twrt}}(r, o)\| > 1$ **then**
42: $\qquad\qquad$ **break** *# Avoid multi-valued hop*
43: $\qquad\quad$ **else if** $e_{\text{ce}} = o$ **and** $\|f_{\text{twhr}}(s, r)\| > 1$ **then**
44: $\qquad\qquad$ **break** *# Avoid multi-valued hop*
45: $\qquad\quad C.\text{append}(t)$
46: $\quad$ **if** $\text{any}([t_0$ **in** $C$ **for** $C$ **in** $\mathcal{T}])$ **then**
47: $\qquad$ **break** *# $t_0$ should in $\mathcal{T}$*
48: *# Reverse the order of triples in each chain*
49: $\mathcal{T} = [C.\text{reverse}()$ **for** $C$ **in** $\mathcal{T}]$
50: **return** $\mathcal{T}$

---

**Step 4. Generality and Locality Subgraphs Sampling:** After obtaining the edit triples, we sample subgraphs for generality and locality. In this work, for simplicity, we restrict the subgraph class to simple chain(s). The distinction between generality and locality lies in whether their structure include the entire edit triple $t_\varepsilon = (s_\varepsilon, r_\varepsilon, o_\varepsilon)$. Therefore, for generality, sampling starts with the $t_\varepsilon$. For locality, there are four possible options: the head entity $s_\varepsilon$, the relation $r_\varepsilon$, the tail $o_\varepsilon$ (only for $o_\varepsilon \in \tilde{E}$), and a random entity $\tilde{e} = \sigma(\tilde{E}, \mathcal{U})$, where $\tilde{E}$ denotes the full set of entities after filtering. Given these options $\{s_\varepsilon, r_\varepsilon, o_\varepsilon, \tilde{e}\}$, the initial triple $t_l$ is generated as:

$$t_l = \begin{cases} \sigma(f_{\text{twr}}(x), \mathcal{U}), & \text{if } x = r_\varepsilon, \\ \sigma(f_{\text{tw}^*}(x), \mathcal{U}), & \text{else.} \end{cases} \quad \text{s.t. } x = \sigma\left(\{s_\varepsilon, o_\varepsilon, r_\varepsilon, \tilde{e}\}, \mathcal{U}\right), \quad f_{\text{tw}^*} = \sigma\left(\{f_{\text{twh}}, f_{\text{twt}}\}, \mathcal{U}\right) \quad (3)$$

where $f_{\text{twr}}(x), f_{\text{twt}}(x)$ denote the functions that retrieve all triples in which $x$ appears as the relation or the tail entity, respectively. These triples are obtained by retrieving the claims fields of all entities in $\tilde{E}$ that contain the ID of $x$, using the search engine. If $t_l = t_\varepsilon$, resampling will be performed.

Then, we uniformly apply NMCS (Algorithm 1) to obtain multi-hop reasoning chains containing the initial triple. For generality, it is $\mathcal{T}_g = \text{NMCS}(t_\varepsilon, \emptyset, 3, 4, \tilde{E})$; for locality, it is $\mathcal{T}_l = \text{NMCS}(t_l, \{t_\varepsilon\}, 3, 4, \tilde{E})$. In this algorithm, $f_{\text{twrt}}(r, o)$ denotes the set of all triples where $r$ is the

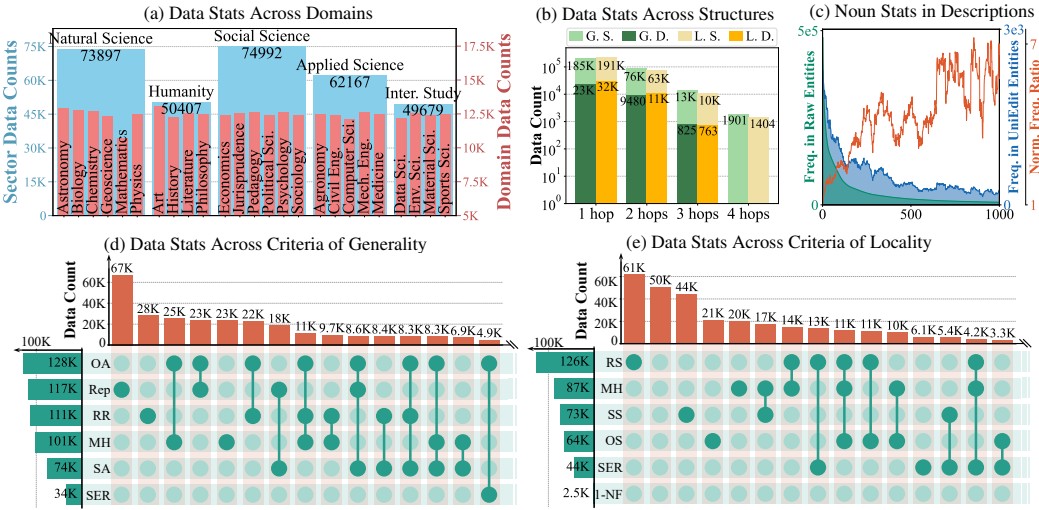

Figure 3: Data count statistics of UNIEDIT across: **(a)** domains, **(b)** multi-hop counts and query chain structures (G., L., S., and D. represent generality, locality, single, and double, respectively), and **(d, e)** the top 15 combinations of recognized evaluation criteria. **(c)** displays the frequency statistics of nouns in entity descriptions.

relation and $o$ is the tail. Similarly, $f_{twhr}(s, r)$ denotes the set of all triples where $s$ is the head and $r$ is the relation. The algorithm is divided into two parts. In the first part (lines 1-23), NMCS samples triples around the initial triple to construct a chain (without considering the prediction object yet). In the second part (lines 24-47), it selects a node $e$ in the chain as the prediction object and expands from both sides (or one side if $e$ is an endpoint in that chain) to form multi-hop chains pointing to $e$, as shown in step 4 of Figure 2. Noting that in lines 41 to 44, NMCS prevents intermediate hops to non-object nodes from being multi-valued, thereby maintaining the clarity of the multi-hop prompt. Through the above process, NMCS uniformly incorporates all structure-related criteria mentioned in Table 1, as well as potential combinations, including MH, RR, SER, and 1-NF.

**Step 5. Final Data Generation:** We utilize Deepseek-V3 [48] to convert the sampled structured data of edit, generality, and locality into natural language to form the final dataset. For each multi-hop sample, we first have it generate single-hop sentences for each triple, and then merge them. To ensure data quality, we conduct automated checks to confirm that each generated prompt contains the subject and correctly points to the object, followed by human evaluation.

## 4.2 Dataset Statistics

UNIEDIT comprises 311K entries, each containing an editing sample, a generality sample, and a locality sample. Figure 3 summarizes key statistics of the dataset, highlighting its broad domain coverage and structural diversity. Figure 3c reports the noun frequency distribution in entity descriptions from both raw Wikidata and UNIEDIT, exhibiting a clear long-tail pattern. The normalized frequency ratio (UNIEDIT/raw) demonstrates that our data construction process effectively reduces the long-tail effect, leading to a more balanced distribution. See Appendix C for more details.

## 5 Experiments

In this section, we conduct a comprehensive evaluation of multiple backbones and editors based on the characteristics of UNIEDIT. This evaluation spans various domains and editing evaluation criteria. Additionally, we assess the domain generalization of the editor that relies on edit training, emphasizing the importance of open-domain datasets in enhancing these editors. Implementation details and additional experiments, including sequential editing and instance analysis, can be found in Appendix D.

Table 2: Overall editing performance on UNIEDIT, with "W/O" indicating results of pre-edit LLMs. "Rel.", "Gen.", and "Loc." are the abbreviations of reliability, generality, and locality, respectively.

| Editors | GPT2-XL (1.5B) | | | | GPT-J (6B) | | | | LlaMa-3.1 (8B) | | | |
|---|---|---|---|---|---|---|---|---|---|---|---|---|
| | Rel. | Gen. | Loc. | Average | Rel. | Gen. | Loc. | Average | Rel. | Gen. | Loc. | Average |
| W/O | 29.69 | 28.04 | **100.0** | $52.58_{\pm0.05}$ | 35.34 | 33.04 | **100.0** | $56.13_{\pm0.03}$ | 43.68 | 51.81 | **100.0** | $65.16_{\pm0.02}$ |
| FT | **100.0** | 49.46 | 89.72 | $79.73_{\pm0.07}$ | **100.0** | 57.25 | 91.26 | $82.84_{\pm0.24}$ | **100.0** | 69.00 | 93.54 | $87.51_{\pm0.17}$ |
| IKE [44] | 99.93 | 76.46 | 83.35 | $86.58_{\pm0.12}$ | 99.80 | 79.05 | 84.31 | $87.72_{\pm0.20}$ | 93.54 | **89.52** | 80.79 | $87.95_{\pm0.30}$ |
| ROME [16] | 92.02 | 35.84 | 96.76 | $74.87_{\pm0.17}$ | 98.98 | 45.33 | 96.41 | $80.24_{\pm0.05}$ | 75.81 | 51.38 | 95.12 | $74.10_{\pm0.13}$ |
| SERAC [37] | 99.46 | **78.79** | 88.06 | $\mathbf{88.77}_{\pm0.10}$ | 99.16 | **81.32** | 86.59 | $\mathbf{89.02}_{\pm0.17}$ | 98.96 | 83.66 | 84.25 | $\mathbf{88.96}_{\pm0.08}$ |
| T-Patcher [38] | 82.28 | 45.40 | 97.27 | $74.98_{\pm0.21}$ | 91.24 | 48.16 | 93.23 | $77.54_{\pm0.33}$ | 73.03 | 49.83 | 83.27 | $68.71_{\pm0.20}$ |
| GRACE [39] | 99.68 | 28.00 | 99.99 | $75.89_{\pm0.03}$ | 99.99 | 33.16 | 99.97 | $77.71_{\pm0.05}$ | 99.92 | 51.89 | 99.97 | $83.93_{\pm0.11}$ |
| AlphaEdit [33] | 92.26 | 37.20 | 95.90 | $75.12_{\pm0.30}$ | 99.77 | 43.91 | 97.60 | $80.43_{\pm0.31}$ | 84.09 | 55.10 | 98.72 | $79.30_{\pm0.24}$ |

## 5.1 Experiments Settings

**LLM Backbones:** We considered backbones of varying sizes and architectures, selecting GPT2-XL (1.5B), GPT-J (6B), and LLaMa-3.1 (8B).

**Editors:** We evaluate various types of editors, spanning methods that modify model parameters or use external modules: Fine-Tuning (FT), ROME [16], AlphaEdit [33], SERAC [37], T-Patcher [38], and GRACE [39]. Additionally, we include IKE [44], which applies in-context learning for model response correction, as a baseline.

## 5.2 Overall Performance

**Editors Struggle with the Challenging Generality Evaluation of UNIEDIT.** Table 2 presents the overall performance. Since the domain knowledge usually follows a long tail distribution (Figure 3c), pre-edit LLMs perform poorly. After editing, most editors effectively guide LLMs to incorporate the intended edits, resulting in high reliability. In particular, FT tends to overfit individual knowledge samples, achieving a perfect reliability score of 100 across all three backbones.

However, editors generally struggle with the more challenging generality evaluation in UNIEDIT. This is especially evident for L&E-based methods such as ROME and AlphaEdit, despite their reported success with simple rephrases in their papers. Similar issues arise with methods like T-Patcher and GRACE. These approaches all do direct backpropagation through edit statements, but often overlook how well the LLM can apply the injected knowledge in broader contexts. IKE and SERAC achieve the best generality performance, leveraging priors learned through in-context learning and edit training. However, giving too much weight to the prior leads to relatively lower locality scores. Notably, GRACE, through its token-based linear distance retrieval mechanism, avoids interference from unrelated samples and thereby results in high locality. Nevertheless, its strong assumption of linear semantic structure in the representation space significantly limits its ability to generalize edits.

## 5.3 Performance Across Domains

Figure 4 illustrates the editing performance of various editors on UNIEDIT across different domains. We evaluated all metrics except MH and SER as these two involve subtle reasoning and are not straightforward for analysis.

Editor performance on reliability shows little variation across domains. However, regarding generality, all editors exhibit a relatively consistent performance distribution: higher scores in Natural Sciences and Humanities, and lower scores in Social Sciences and Applied Sciences. We hypothesize that this stems from the distributional bias of LLMs' pretraining corpora, which enables better generalization for incorporated knowledge in well-represented domains.

For locality, the performance distribution over domains from different editors is less consistent. However, all editors achieve a relatively high score on Humanities. We attribute this to the robustness gained from the models' greater exposure to literary content during pretraining. These observations underscore the importance of open-domain knowledge editing, particularly for underrepresented or low-resource domains that receive less attention in existing pretraining corpora and should be prioritized in future research.

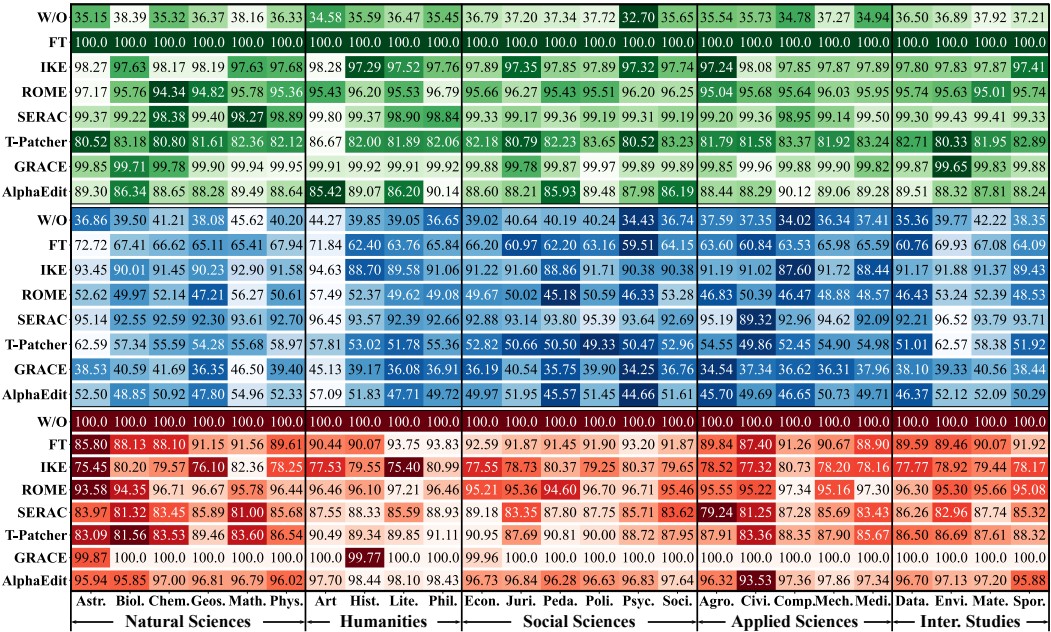

| | Astr. | Biol. | Chem. | Geos. | Math. | Phys. | Art | Hist. | Lite. | Phil. | Econ. | Juri. | Peda. | Poli. | Psyc. | Soci. | Agro. | Civi. | Comp. | Mech. | Medi. | Data. | Envi. | Mate. | Spor. |
|---|---|---|---|---|---|---|---|---|---|---|---|---|---|---|---|---|---|---|---|---|---|---|---|---|---|
| W/O | 35.15 | 38.39 | 35.32 | 36.37 | 38.16 | 36.33 | 34.58 | 35.59 | 36.47 | 35.45 | 36.79 | 37.20 | 37.34 | 37.72 | 32.70 | 35.65 | 35.54 | 35.73 | 34.78 | 37.27 | 34.94 | 36.50 | 36.89 | 37.92 | 37.21 |
| FT | 100.0 | 100.0 | 100.0 | 100.0 | 100.0 | 100.0 | 100.0 | 100.0 | 100.0 | 100.0 | 100.0 | 100.0 | 100.0 | 100.0 | 100.0 | 100.0 | 100.0 | 100.0 | 100.0 | 100.0 | 100.0 | 100.0 | 100.0 | 100.0 | 100.0 |
| IKE | 98.27 | 97.63 | 98.17 | 98.19 | 97.63 | 97.68 | 98.28 | 97.29 | 97.52 | 97.76 | 97.89 | 97.35 | 97.85 | 97.89 | 97.32 | 97.74 | 97.24 | 98.08 | 97.85 | 97.87 | 97.89 | 97.80 | 97.83 | 97.87 | 97.41 |
| ROME | 97.17 | 95.76 | 94.34 | 94.82 | 95.78 | 95.36 | 95.43 | 96.20 | 95.53 | 96.79 | 95.66 | 96.27 | 95.43 | 95.51 | 96.20 | 96.25 | 95.04 | 95.68 | 95.64 | 96.03 | 95.95 | 95.74 | 95.63 | 95.01 | 95.74 |
| SERAC | 99.37 | 99.22 | 98.38 | 99.40 | 98.27 | 98.89 | 99.80 | 99.37 | 98.90 | 98.84 | 99.33 | 99.17 | 99.36 | 99.19 | 99.31 | 99.19 | 99.20 | 99.36 | 98.95 | 99.14 | 99.50 | 99.30 | 99.43 | 99.41 | 99.33 |
| T-Patcher | 80.52 | 83.18 | 80.80 | 81.61 | 82.36 | 82.12 | 86.67 | 82.00 | 81.89 | 82.06 | 82.18 | 80.79 | 82.23 | 83.65 | 80.52 | 83.23 | 81.79 | 81.58 | 83.37 | 81.92 | 83.24 | 82.71 | 80.33 | 81.95 | 82.89 |
| GRACE | 99.85 | 99.71 | 99.78 | 99.90 | 99.94 | 99.95 | 99.91 | 99.92 | 99.91 | 99.92 | 99.88 | 99.78 | 99.87 | 99.97 | 99.89 | 99.89 | 99.85 | 99.96 | 99.88 | 99.90 | 99.82 | 99.87 | 99.65 | 99.83 | 99.88 |
| AlphaEdit | 89.30 | 86.34 | 88.65 | 88.28 | 89.49 | 88.64 | 85.42 | 89.07 | 86.20 | 90.14 | 88.60 | 88.21 | 85.93 | 89.48 | 87.98 | 86.19 | 88.44 | 88.29 | 90.12 | 89.06 | 89.28 | 89.51 | 88.32 | 87.81 | 88.24 |
| W/O | 36.86 | 39.50 | 41.21 | 38.08 | 45.62 | 40.20 | 44.27 | 39.85 | 39.05 | 36.65 | 39.02 | 40.64 | 40.19 | 40.24 | 34.43 | 36.74 | 37.59 | 37.35 | 34.02 | 36.34 | 37.41 | 35.36 | 39.77 | 42.22 | 38.35 |
| FT | 72.72 | 67.41 | 66.62 | 65.11 | 65.41 | 67.94 | 71.84 | 62.40 | 63.76 | 65.84 | 66.20 | 60.97 | 62.20 | 63.16 | 59.51 | 64.15 | 63.60 | 60.84 | 63.53 | 65.98 | 65.59 | 60.76 | 69.93 | 67.08 | 64.09 |
| IKE | 93.45 | 90.01 | 91.45 | 90.23 | 92.90 | 91.58 | 94.63 | 88.70 | 89.58 | 91.06 | 91.22 | 91.60 | 88.86 | 91.71 | 90.38 | 90.38 | 91.19 | 91.02 | 87.60 | 91.72 | 88.44 | 91.17 | 91.88 | 91.37 | 89.43 |
| ROME | 52.62 | 49.97 | 52.14 | 47.21 | 56.27 | 50.61 | 57.49 | 52.37 | 49.62 | 49.08 | 49.67 | 50.02 | 45.18 | 50.59 | 46.33 | 53.28 | 46.83 | 50.39 | 46.47 | 48.88 | 48.57 | 46.43 | 53.24 | 52.39 | 48.53 |
| SERAC | 95.14 | 92.55 | 92.59 | 92.30 | 93.61 | 92.70 | 96.45 | 93.57 | 92.39 | 92.66 | 92.88 | 93.41 | 93.80 | 95.39 | 93.64 | 92.69 | 95.19 | 89.32 | 92.96 | 94.62 | 92.09 | 92.21 | 96.52 | 93.79 | 93.71 |
| T-Patcher | 62.59 | 57.34 | 55.59 | 54.28 | 55.68 | 58.97 | 57.81 | 53.02 | 51.78 | 55.36 | 52.82 | 50.66 | 50.50 | 49.33 | 50.47 | 52.96 | 54.55 | 49.86 | 52.45 | 54.90 | 54.98 | 51.01 | 52.57 | 58.38 | 51.92 |
| GRACE | 38.53 | 40.59 | 41.69 | 36.35 | 46.50 | 39.40 | 45.13 | 39.17 | 36.08 | 36.91 | 36.19 | 40.54 | 35.75 | 39.90 | 34.25 | 36.76 | 34.54 | 37.34 | 36.62 | 36.31 | 37.96 | 38.10 | 39.33 | 40.56 | 38.44 |
| AlphaEdit | 52.50 | 48.85 | 50.92 | 47.80 | 54.96 | 52.33 | 57.09 | 51.83 | 47.71 | 49.72 | 49.97 | 51.95 | 45.57 | 51.45 | 44.66 | 51.61 | 45.70 | 49.69 | 46.65 | 50.73 | 49.71 | 46.37 | 52.12 | 52.09 | 50.29 |
| W/O | 100.0 | 100.0 | 100.0 | 100.0 | 100.0 | 100.0 | 100.0 | 100.0 | 100.0 | 100.0 | 100.0 | 100.0 | 100.0 | 100.0 | 100.0 | 100.0 | 100.0 | 100.0 | 100.0 | 100.0 | 100.0 | 100.0 | 100.0 | 100.0 | 100.0 |
| FT | 85.80 | 88.13 | 88.10 | 91.15 | 91.56 | 89.61 | 90.44 | 90.07 | 93.75 | 93.83 | 92.59 | 91.87 | 91.45 | 91.90 | 93.20 | 91.87 | 89.84 | 87.40 | 91.26 | 90.67 | 88.90 | 89.59 | 89.46 | 90.07 | 91.92 |
| IKE | 75.45 | 80.20 | 79.57 | 76.10 | 82.36 | 78.25 | 77.53 | 79.55 | 75.40 | 80.99 | 77.55 | 78.73 | 80.37 | 79.25 | 80.37 | 79.65 | 78.52 | 77.32 | 80.73 | 78.20 | 78.16 | 77.77 | 78.92 | 79.44 | 78.17 |
| ROME | 93.58 | 94.35 | 96.71 | 96.67 | 95.78 | 96.44 | 96.46 | 96.10 | 97.21 | 96.46 | 95.21 | 95.36 | 94.60 | 96.70 | 96.71 | 95.46 | 95.55 | 95.22 | 97.34 | 95.16 | 97.30 | 96.30 | 95.30 | 95.66 | 95.08 |
| SERAC | 83.97 | 81.32 | 83.45 | 85.89 | 81.00 | 85.68 | 87.55 | 88.33 | 85.59 | 88.93 | 89.18 | 87.80 | 87.75 | 85.71 | 83.62 | | 79.24 | 81.25 | 87.28 | 85.69 | 83.43 | 86.26 | 82.96 | 87.74 | 85.32 |
| T-Patcher | 83.09 | 81.56 | 83.53 | 89.46 | 83.60 | 86.54 | 90.49 | 89.34 | 89.85 | 91.11 | 90.95 | 87.69 | 90.81 | 90.00 | 88.72 | 87.95 | 87.91 | 83.36 | 88.35 | 87.90 | 85.67 | 86.50 | 86.69 | 87.61 | 88.32 |
| GRACE | 99.87 | 100.0 | 100.0 | 100.0 | 100.0 | 100.0 | 100.0 | 99.77 | 100.0 | 100.0 | 99.96 | 100.0 | 100.0 | 100.0 | 100.0 | 100.0 | 100.0 | 100.0 | 100.0 | 100.0 | 100.0 | 100.0 | 100.0 | 100.0 | 100.0 |
| AlphaEdit | 95.94 | 95.85 | 97.00 | 96.81 | 96.79 | 96.02 | 97.70 | 98.44 | 98.10 | 98.43 | 96.73 | 96.84 | 96.28 | 96.63 | 96.83 | 97.64 | 96.32 | 93.53 | 97.36 | 97.86 | 97.34 | 96.70 | 97.13 | 97.20 | 95.88 |
| | **Natural Sciences** | | | | | | **Humanities** | | | | **Social Sciences** | | | | | | **Applied Sciences** | | | | | **Inter. Studies** | | | |

Figure 4: Editing performance on UNIEDIT across domains, with each metric representing the average result across three post-edit backbones. The color bands (top to bottom) indicate reliability (green), generality (blue), and locality (red), with ranges normalized across domains (rows).

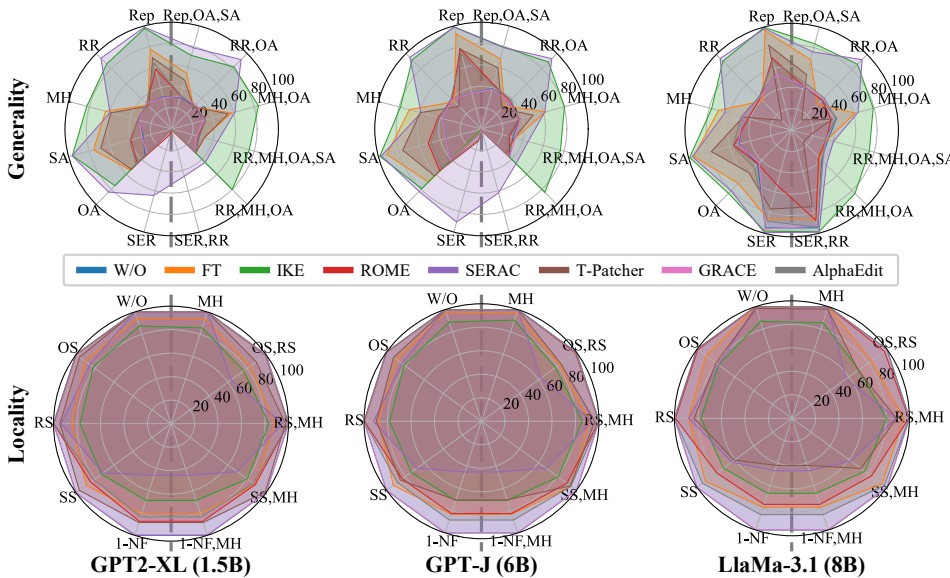

Figure 5: Editing performance across combinations of generality and locality evaluation criteria. The left half of each radar chart shows the evaluation results for a single criterion, while the symmetrical right half reflects the results after combining it with others.

## 5.4 Performance Across Evaluation Criteria

Figure 5 shows the generality and locality results across various criteria combinations. For generality, most editors score lower on more complex evaluation, such as the comparison between Rep and the combination of Rep, OA, and SA, or the results of SA versus RR, MH, OA, and SA. This occurs because the edit information is part of a natural language sentence covering multiple evaluation criteria. The more intricate the structure, the harder it is for the edited knowledge to be recognized and applied. Exceptions exist, such as IKE's performance on OA and the combination of RR, MH, and OA. We attribute this to the higher frequency of the combination compared to standalone OA in UNIEDIT (Figure 3d), leading to a sampling bias in the demonstrations for in-context learning.

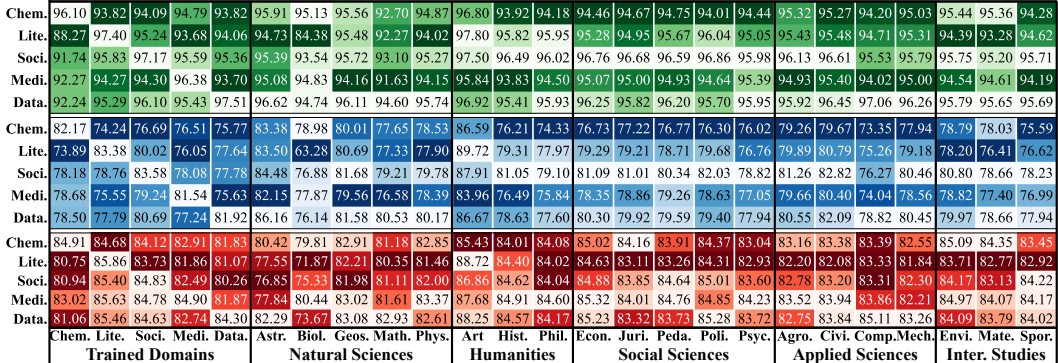

Figure 6: Editing performance of SERAC trained on five domains from different sectors in UniEdit, using GPT2-XL as the backbone. The color bands (top to bottom) represent reliability (green), generality (blue), and locality (red), with ranges normalized across domains (columns).

However, for locality, adding MH to the locality evaluation does not lead to a performance decline compared to the single criterion. In some cases, performance even improves, as seen in the results of SS versus the combination of SS and MH. This contrasts with the findings for generality, but we believe it stems from the same underlying principle: complex sentences reduce the likelihood of overlapping components between locality inputs and the edited knowledge, preventing interference with the model's original response. An exception is the combination of OS and RS, which creates dual overlap with the edit sample, making the evaluation more challenging than a standalone OS. In general, the addition of complexities increases the challenge of generality much more than locality.

## 5.5 Domain Generalization of Edit Training

To evaluate the impact of open-domain knowledge on edit training, we assess the editing performance of SERAC trained on five domains from different sectors, as shown in Figure 6. The first five columns clearly show that training in a specific domain results in better performance when tested on the corresponding domain. In terms of reliability and generality, it can be observed that similar or overlapping training and testing domains tend to yield better results, such as the performance in Biology when trained on Chemistry, or Computer Science when trained on Data Science. For locality, due to the limited relevance of these samples to each domain (usually only a small portion of domain-specific elements), the results across different training domains show minimal variation.

Additionally, compared to Figure 4, the editing performance of SERAC, particularly concerning generality, decreases significantly. This analysis suggests that the scale and breadth of the training data significantly influence the effectiveness of edit training-based editors.

## 6 Conclusion, Limitation and Future Works

We construct a open-domain LLM knowledge editing benchmark, UniEdit. By introducing a unified NMCS algorithm, we integrate most existing evaluation criteria and induce potential composite patterns, thereby posing greater challenges for editing evaluation. We conduct extensive analyses across multiple editors and backbones on UniEdit, with key findings as follows: (1) Editors, especially those following the L&E paradigm, show notable limitations in handling complex generality. (2) Performance varies across domains, underscoring the importance of low-resource knowledge editing. (3) Higher sample complexity increases generality difficulty, but may ease locality evaluation. (4) The scale and domain coverage of training data affect the performance of editors that rely on editing training. Regarding the limitations, UniEdit currently focuses on English and lacks evaluations for other languages [49, 50]. Additionally, it emphasizes a single language modality and does not include challenging evaluations for other modals, such as vision LLM editing [51–53]. Therefore, future work could expand research using our toolkit in several ways: (1) Extending benchmarks to include languages beyond English; (2) Leveraging multimodal content from Wikidata (e.g., videos, images) to develop more comprehensive multi-modal editing benchmarks; (3) Exploring more fine-grained, long-tail domains and incorporating more diverse evaluation criteria.

## Acknowledgements

This work is supported by the National Science and Technology Major Project (2022ZD0120302). In addition, we sincerely thank the Shanghai Institute of AI for Education (IAIE) for their computational power support.

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

Table 3: UNIEDIT instances of the generality criteria, with gray arrows in the **Structures** column showing the direction of prompt construction for each criterion example.

| Domains | Edit Samples | | Criteria Instances | Structures |
|---|---|---|---|---|
| **Computer Science** | The port of the Firefox web browser to the AmigaOS 4 platform, known as Timberwolf, was first created in → 2010 AD | **Rep** | The inception of Timberwolf occurred in 2010 AD. | $s_\varepsilon \xrightarrow{r_\varepsilon} o_\varepsilon$ |
| **Chemistry** | The meteorite known as Alkali was discovered in → Nevada (NV, Nevada, United States). | **MH** | The minimum temperature ever recorded in the location where Alkali was discovered is −50 degree Fahrenheit. | $s_\varepsilon \xrightarrow{r_\varepsilon} o_\varepsilon$, $r_A$, $e_A$ |
| **Agronomy** | The subspecies of plant Satureja horvatii subsp. macrophylla has the basionym → Satureja parnassica var. macrophylla. | **RR** | The taxon that has the basionym Satureja parnassica var. macrophylla is Satureja horvatii subsp. macrophylla. | $s_\varepsilon \xrightarrow{r_\varepsilon} o_\varepsilon$ |
| **Political Science** | The book "Rechtsstaat statt Revolution, Verrechtlichung statt Demokratie?", discussing German and Spanish theory of law and political history, was edited by → Frieder Otto Wolf. | **SER** | Is the editor of "Rechtsstaat statt Revolution, Verrechtlichung statt Demokratie?" the same as the editor of "Die Tätigkeit der Philosophen"? Yes. | $s_\varepsilon \xrightarrow{r_\varepsilon} o_\varepsilon$, $r_B$, $e_B$ |
| **Civil Engineering** | Geotechnical engineering (also known as geotechnics) is a specialized branch of → construction engineering. | **SA** | Geotechnics is a subclass of construction engineering. | $s'_\varepsilon \xrightarrow{r_\varepsilon} o_\varepsilon$ |
| **Art** | The musical composition Die Weihnachtsgeschichte was composed by → Hugo Distler (August Hugo Distler). | **OA** | The composer of Die Weihnachtsgeschichte is August Hugo Distler. | $s_\varepsilon \xrightarrow{r_\varepsilon} o'_\varepsilon$ |
| **Medicine** | The genetic variant VHL I180V (c.538A>G) (also known as I180V (c.538A>G) or C.538A>G) is located on → human chromosome 3 (chr3, Homo sapiens chromosome 3). | **MH, RR, SA** | The genetic variant located on the same chromosome as MIR1263 is C.538A>G. | $s'_\varepsilon \xrightarrow{r_\varepsilon} o_\varepsilon$, $r_C$, $e_C$ |

# A   Editing Evaluation Criteria

In this section, we introduce the fine-grained evaluation criteria recognized in existing studies, including generality and locality. Examples illustrating these criteria are presented in Table 3 and Table 4. The following content builds upon the definitions provided in Section 3.

## A.1   Criteria for Generality

**Rephrase (Rep) :** Rep is the most straightforward generality criterion. Given $\mathcal{E} = \{(f_{\text{nl}}(s_\varepsilon, r_\varepsilon), o_\varepsilon)\}$, it examines whether $f'_{\text{llm}}(f'_{\text{nl}}(o_\varepsilon, r_\varepsilon)) = s_\varepsilon$, where $f'_{\text{nl}}$ represents a natural language generation function with a different syntactic structure from $f_{\text{nl}}$.

**Multi-Hop (MH) :** Given $\mathcal{E} = \{(f_{\text{nl}}(s_{\varepsilon_0}, r_{\varepsilon_0}), o_{\varepsilon_0})\} \cup \{(f_{\text{nl}}(o_{\varepsilon_{i-1}}, r_{\varepsilon_i}), o_{\varepsilon_i})\}_{i=1}^{\tau}$, MH examines whether $f'_{\text{llm}}$ can infer the final entity $o_{\varepsilon_\tau}$ based on the initial entity $s_{\varepsilon_0}$ and a sequence of relations, i.e., $f'_{\text{llm}}(f_{\text{nl}}(s_{\varepsilon_0}, r_{\varepsilon_0}, r_{\varepsilon_1}, \ldots)) = o_{\varepsilon_\tau}$.

**Relation Reversal (RR) :** Given $\mathcal{E} = \{(f_{\text{nl}}(s_\varepsilon, r_\varepsilon), o_\varepsilon)\}$, RR examines whether $f'_{\text{llm}}$ can infer the subject $s_\varepsilon$ based on the object $o_\varepsilon$ and the inverse relation $r_\varepsilon$, i.e., $f'_{\text{llm}}(f_{\text{nl}}(o_\varepsilon, r'_\varepsilon)) = s_\varepsilon$.

**Same Entity Recognition (SER) :** Given $\mathcal{E} = \{(f_{\text{nl}}(s_{\varepsilon_1}, r_{\varepsilon_1}), o_{\varepsilon_1}), (f_{\text{nl}}(s_{\varepsilon_2}, r_{\varepsilon_2}), o_{\varepsilon_1})\}$, SER evaluates whether $f'_{\text{llm}}$ can correctly determine that $f_{\text{nl}}(o_{\varepsilon_1}, r_{\varepsilon_1})$ and $f_{\text{nl}}(o_{\varepsilon_2}, r_{\varepsilon_2})$ refer to the same entity, where the two prompts are merged into a single judgment question. As shown in Figure 1, the structured data corresponds to either a single chain or a double chain. Only the double chain is used to generate data containing SER.

Table 4: UNIEDIT instances of the locality criteria, with gray arrows in the **Structures** column showing the direction of prompt construction for each criterion example.

| Domains | Edit Samples | | Criteria Instances | Structures |
|---|---|---|---|---|
| **Mathematics** | A graded Lie algebra, which is a Lie algebra equipped with a grading compatible with the Lie bracket, is defined by the $\mathfrak{g} = \bigoplus_{i \in \mathbb{Z}} \mathfrak{g}_i$ formula $\rightarrow [-,-] \colon \mathfrak{g}_i \otimes \mathfrak{g}_j \rightarrow \mathfrak{g}_{i+j}$. | W/O | The width of the artwork depicting the marriage of the archduke Maximilian of Austria and the duchess Mary of Burgundy, created in 1635, is 175 centimeters. | $e_D \xrightarrow{r_E} e_E$ |
| **History** | Dou Rong, a high minister during the early decades of the Later Han period, was given the posthumous name $\rightarrow$ 戴 (Dai). | SS | The sibling of Dou Rong is Dou You. | $s_\varepsilon \xrightarrow{r_F} e_F$ |
| **Literature** | The poetry collection Erlösungen, which contains autobiographical references by Richard Dehmel, is dedicated to $\rightarrow$ Friedrich Nietzsche. | RS | Toyagasaki-jinja is dedicated to Toyotama-hime. | $e_G \xrightarrow{r_\varepsilon} e_H$ |
| **Geoscience** | The Neodani Fault in Japan was caused by $\rightarrow$ 1891 Nōbi earthquake (Nobi earthquake). | OS | The coordinate location of the 1891 Nōbi earthquake is Earth: latitude 35.60, longitude 136.30. | $o_\varepsilon \xrightarrow{r_I} e_I$ |
| **Biology** | The cell type known as transitional B cell (also referred to as Transitional B cell) was discovered by $\rightarrow$ David Allman (researcher, ORCID 0000-0003-2732-2686). | 1-NF | The cell type discovered or invented by Michael P Cancro is transitional B cell. | $s_\varepsilon \xrightarrow{r_\varepsilon} e_J$ |
| **Astronomy** | The diameter of the Helen Sawyer Hogg Telescope (also known as HSHT or CASLEO:HSHT) is $\rightarrow$ 0.61 metre. | MH, SS | The asteroid discovered at the astronomical complex that includes Helen Sawyer Hogg Telescope is 2189 Zaragoza. | $e_K \xrightarrow{r_K} e_L$ ; $r_L$ ; $s_\varepsilon$ |

**Subject Alias (SA)** : Given $\mathcal{E} = \{(f_{\text{nl}}(s_\varepsilon, r_\varepsilon), o_\varepsilon)\}$, SA assesses whether $f'_{\text{llm}}$ can effectively recognize a subject alias $s'_\varepsilon$ and produce the correct response, i.e., $f'_{\text{llm}}(f_{\text{nl}}(s'_\varepsilon, r_\varepsilon)) = o_\varepsilon$.

**Object Alias (OA):** Given $\mathcal{E} = \{(f_{\text{nl}}(s_\varepsilon, r_\varepsilon), o_\varepsilon)\}$, OA assesses whether $f'_{\text{llm}}$ can predict an alias of the object, $o'_\varepsilon$, i.e., $f'_{\text{llm}}(f_{\text{nl}}(s_\varepsilon, r_\varepsilon)) = o'_\varepsilon$. In practice, we take the top-k predicted tokens to evaluate its recall of the corresponding token sequence.

## A.2 Criteria for Locality

The definition of locality criteria is relatively simpler compared to generality. The challenge of locality mainly arises from its overlap with the edit triple $t_\varepsilon = (s_\varepsilon, r_\varepsilon, o_\varepsilon)$. Below, we define each criterion and align it with the structured definitions illustrated in Figure 1.

**Completely unrelated (W/O):** W/O expects $f'_{\text{llm}}$ to preserve responses to facts that are completely unrelated to $t_\varepsilon$, i.e., $f'_{\text{llm}}(f_{\text{nl}}(s, r)) = f_{\text{llm}}(f_{\text{nl}}(s, r))$, where $\{s\} \cap \{s_\varepsilon, o_\varepsilon\} = \emptyset$ and $r \neq r_\varepsilon$. It corresponds to the non-crossed case.

**Subject Specificity (SS)** : SS expects $f'_{\text{llm}}$ to preserve responses to facts related to $s_\varepsilon$, i.e., $f'_{\text{llm}}(f_{\text{nl}}(s_\varepsilon, r)) = f_{\text{llm}}(f_{\text{nl}}(s_\varepsilon, r))$ holds for $r \neq r_\varepsilon$. It corresponds to the subject-crossed case.

**Relation Specificity (RS)** : RS expects $f'_{\text{llm}}$ to preserve responses to facts related to $r_\varepsilon$, i.e., $f'_{\text{llm}}(f_{\text{nl}}(s, r_\varepsilon)) = f_{\text{llm}}(f_{\text{nl}}(s, r_\varepsilon))$ holds for $\{s\} \cap \{s_\varepsilon, o_\varepsilon\} = \emptyset$. It corresponds to the relation-crossed case.

**Object Specificity (OS):** OS expects $f'_{\text{llm}}$ to preserve responses to facts related to $o_\varepsilon$, i.e., $f'_{\text{llm}}(f_{\text{nl}}(o_\varepsilon, r)) = f_{\text{llm}}(f_{\text{nl}}(o_\varepsilon, r))$ holds for $r \neq r_\varepsilon$. It corresponds to the object-crossed case.

**1-N Forgotten (1-NF)** : For a one-to-many relation $r_\varepsilon$, 1-NF expects $f'_{\text{llm}}(f_{\text{nl}\setminus o_\varepsilon}(s_\varepsilon, r_\varepsilon)) = f_{\text{llm}}(f_{\text{nl}\setminus o_\varepsilon}(s_\varepsilon, r_\varepsilon))$, where $f_{\text{nl}\setminus o_\varepsilon}(s_\varepsilon, r_\varepsilon)$ prompts the LLM to recall objects excluding $o_\varepsilon$. It corresponds to the subject-relation-crossed case. In addition, since subject and object are symmetrical, NMCS also accordingly introduces object-relation-crossed case, which can be formulated as

Table 5: Partial keywords of each domain and count of retrieved entities.

| Sectors | Domains | Keywords | # Entities |
|---------|---------|----------|-----------|
| Nat. Sci. | Astronomy | constellation, dark energy, radiation, cosmological | 557,136 |
| | Biology | phylogeny, reproductive, ecological, vaccination | 4,966,158 |
| | Chemistry | nanotechnology, molecular, ionic, polymer, pH | 1,606,057 |
| | Geoscience | fossil, glacier, volcanology, erosional, lava, sediment | 1,051,126 |
| | Mathematics | vector space, proof, trigonometry, algebra, continuity | 866,576 |
| | Physics | radiation, quantum, dark energy, velocity, relativity | 249,085 |
| Human. | Art | rhythm, painting, figurative, artwork, artist, gallery | 2,882,212 |
| | History | conquest, biography, monarchy, chronicle, dictatorship | 1,734,319 |
| | Literature | figurative, biography, poetry, metaphorical, emotional | 864,289 |
| | Philosophy | analytic, objective, universal, idealism, atheistic | 176,704 |
| Soc. Sci. | Economics | market, economical, global, developmental, economic | 424,523 |
| | Jurisprudence | international law, administrative law, dispute, tribunal | 471,473 |
| | Pedagogy | inclusive education, syllabus, curricular, discipline | 300,350 |
| | Political Science | ideology, electoral system, political party, socialism | 1,783,002 |
| | Psychology | behavioral, depressed, emotional, empathy, anxious | 587,128 |
| | Sociology | inequality, public policy, racial, collective behavior | 1,049,245 |
| App. Sci. | Agronomy | hydroponics, irrigated, agroforestry, ecological | 720,670 |
| | Civil Engineering | sustainable, construction site, earthquake-resistant | 982,906 |
| | Computer Science | server, database, binary, debugged, version control | 877,716 |
| | Mechanical Engineering | casting, pulley, manufacturing, shaft, cylinder, valve | 230,953 |
| | Medicine | disease, surgery, palliative, therapy, postoperative | 700,260 |
| Inter. Stu. | Data Science | random forest, preprocessed, supervised learning | 113,383 |
| | Environmental Science | environmental impact, contamination, weather-related | 3,344,141 |
| | Material Science | ductility, material processing, bio-compatible | 200,031 |
| | Sports Science | exercise, hydrated, rehabilitation, muscle, workout | 964,996 |

$f'_{\text{llm}}(f_{\text{nl}\setminus s_\varepsilon}(o_\varepsilon, r'_\varepsilon)) = f_{\text{llm}}(f_{\text{nl}\setminus s_\varepsilon}(o_\varepsilon, r'_\varepsilon))$, where $r'_\varepsilon$ denotes the inverse of $r_\varepsilon$ and also follows a one-to-many mapping.

The above definitions cover only the individual criteria. Based on NMCS and the random selection of aliases, various combinations of these criteria can form highly challenging and comprehensive editing evaluations, as illustrated by the example in the bottom row of Table 3 and Table 4.

# B  Construction Details of UNIEDIT

The detail of the enumeration of domain keywords (step 2) is shown below, and the prompts used for final data generation (step 5) are provided in Appendix E.

**Enumeration of Domain Keywords (Step 2):** We use the following prompt to generate domain keywords with GPT-4:

---

**Prompt to Generate Domain Keywords**

Provide the derivative vocabulary and terminology for the domain, including nouns and adjectives. For example, in the subject of biology, words like biologist, organism, and cell are included. Pay attention to the polysemy of words, such as 'abstract' and 'traditional' in the art subject, which can cause confusion, so do not include them. Enclose each word in double quotation marks and separate them with commas. Use an additional line break to separate nouns and adjectives. Be careful not to generate same words repeatedly.
Now, provide the derivative vocabulary for the domain of {}:

---

Approximately 100 keywords are generated for each domain, with Table 5 showing part of keywords from each domain.

Table 6: Data count statistics of UNIEDIT across different data types. The right six columns show the counts of non-entity tails, with "Coord." and "MNLT" representing globe-coordinate and monolingual text, respectively.

| Types | Data | Entity | Relation | String | Quantity | Time | Math | Coord. | MNLT |
|---|---|---|---|---|---|---|---|---|---|
| Edit | 311,142 | 363,014 | 1,770 | 13,434 | 29,211 | 26,669 | 2,377 | 4,940 | 167 |
| Generality | 311,142 | 440,772 | 1,864 | 15,220 | 35,889 | 33,416 | 2,637 | 7,810 | 192 |
| Locality | 311,142 | 394,889 | 1,784 | 16,126 | 31,417 | 31,427 | 1,730 | 19,506 | 128 |
| Union | 933,426 | 703,282 | 1,934 | 44,780 | 96,517 | 91,512 | 6,744 | 32,256 | 487 |

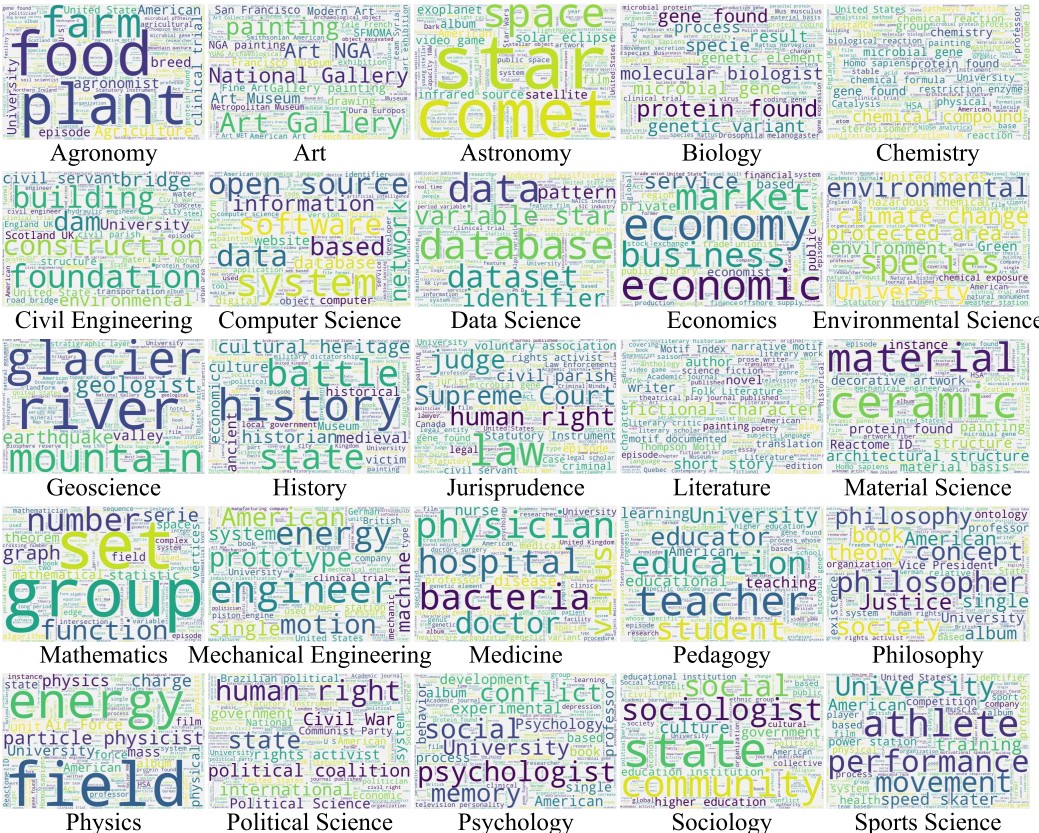

Figure 7: Word cloud of head entity descriptions across domains.

## C  Additional Statistics and Discussion for UNIEDIT

### C.1  Basic Statistics

Table 6 presents the distribution of UNIEDIT across different data types. Figure 7 shows the word cloud distributions of head entity descriptions in the edit samples across different domains. Figure 8 and Figure 9 present the complete data count statistics across combinations of recognized criteria for generality and locality, respectively. These statistics highlight the diversity and comprehensive coverage of UNIEDIT in terms of data types, knowledge domains, and evaluation criteria.

### C.2  Human Assessment of Data Quality

To validate data quality, we expand on the human evaluation process. Based on the sampling theory formula $n = Z^2 p(1-p)/e^2 = 1.96^2 \cdot 0.5 \cdot (1-0.5)/0.05^2 \approx 385$, we randomly select 385 items from UniEdit (containing 311K items, treated as a large population) for human evaluation. This corresponds to a 95% confidence level (reflected by the z-score $Z = 1.96$) and a $\pm 5\%$ margin of

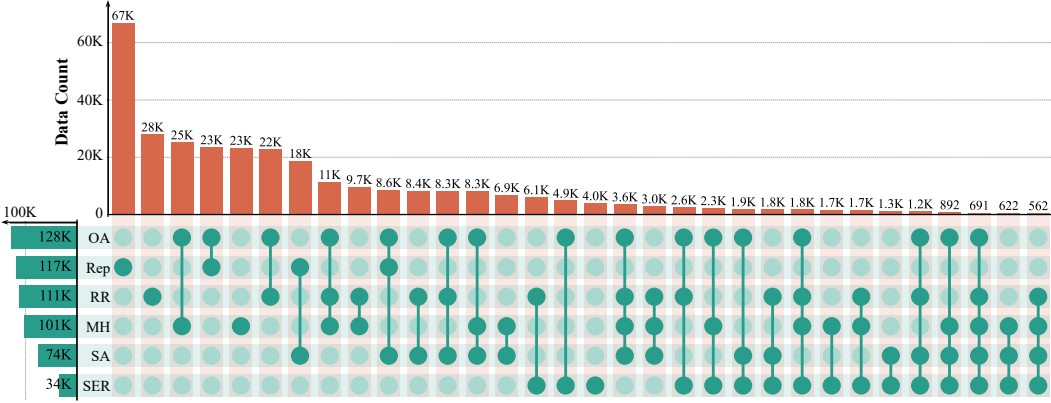

Figure 8: Data count statistics of UNIEDIT across combinations of recognized generality criteria.

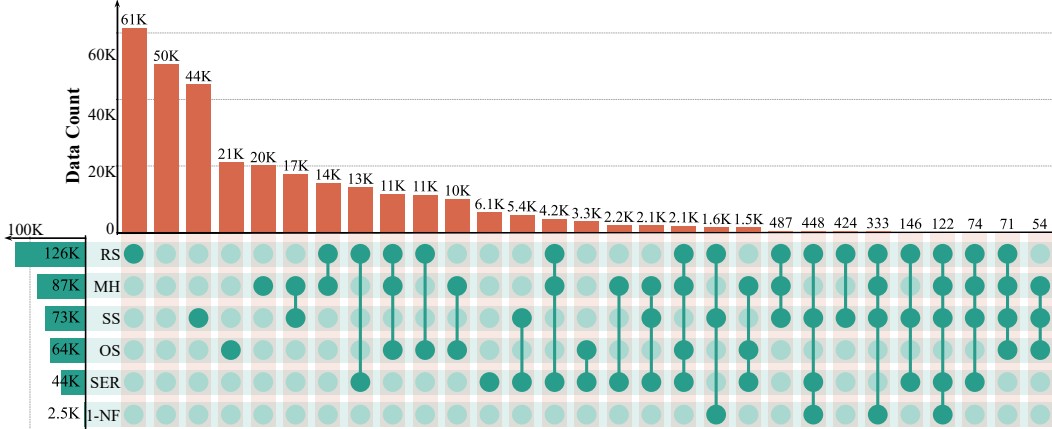

Figure 9: Data count statistics of UNIEDIT across combinations of recognized locality criteria.

error ($e = 0.05$). Here, $p$ denotes the estimated proportion in the population and is set to 0.5 to yield the maximum required sample size. The formula gives the smallest sample size needed to ensure the evaluation reflects the population within an acceptable margin of error.

The human evaluation is conducted according to two criteria:

**Fluency (Score: 1–5):** Measures whether the prompt is grammatically correct and conforms to natural language usage.

1 - severely ungrammatical or unnatural;

3 - generally fluent with minor errors;

5 - fully fluent and natural, with no grammatical issues.

**Logical Consistency (Score: 1–5):** Evaluates whether the generated prompt is logically consistent with the structured multi-hop chain.

1 - the prompt completely fails to represent the multi-hop chain or introduces errors;

3 - partially reflects the reasoning structure but contains missing or ambiguous logic;

5 - fully consistent with the reasoning chain, clearly and faithfully reflecting all steps.

Five researchers independently evaluate the sampled data. We use Krippendorff's alpha to assess the agreement among evaluators. The core idea is to compare the observed disagreement ($D_o$) with the disagreement expected under random assignment ($D_e$), using the formula: $\alpha = 1 - D_o/D_e$. An $\alpha$ of 1 indicates perfect agreement, 0 means agreement at the level of randomness, and negative values indicate systematic disagreement. Krippendorff's alpha supports various levels of measurement,

Table 7: Human-based assessment of UniEdit quality.

| Prompt Type | Criterion | Mean Score | Agreement |
|---|---|---|---|
| Edit Rquest | Fluency | 4.81 | 0.60 |
| | Logical Consistency | 4.92 | 0.46 |
| Generality | Fluency | 4.75 | 0.54 |
| | Logical Consistency | 4.72 | 0.63 |
| Locality | Fluency | 4.78 | 0.61 |
| | Logical Consistency | 4.67 | 0.57 |

including nominal, ordinal, interval, and ratio. We adopt the interval level of measurement to accurately capture the numerical differences between raters' scores. The results of the assessment are presented in the Table 7

### C.3 Discussion on Bias Propagation from Source Data

Benefiting from Wikidata's global crowdsourcing, the platform offers broad knowledge coverage as its primary advantage. However, its open nature also introduces risks of erroneous or inconsistent content, which may propagate into UniEdit. UniEdit is specifically designed to promote editability and logical consistency, allowing models to follow edits while maintaining coherent reasoning chains. Even if some knowledge is factually incorrect, the multi-hop structure of the knowledge graph ensures that inference paths remain logically self-consistent. For example, suppose Wikidata contains an incorrect fact such as (United States, capital, New York). Then, another triple sampled under the generality setting might be (The Statue of Liberty, located in, New York), rather than anything related to Washington, D.C. Given both triples are injected into the LLM, a prompt like "Is the Statue of Liberty located in the capital of the United States?" should yield a positive answer. As such, isolated factual errors have limited impact on the dataset's evaluation reliability. For comparison, the Counterfact [16] dataset is directly built upon counterfactual knowledge.

Another potential issue in Wikidata is the imbalance in attribute richness across entities: mainstream entities tend to have more comprehensive property links, in contrast to less common entities. This may cause UniEdit to support more complex and diverse evaluation criteria for popular entities, while rare ones may only allow for simpler criteria, such as rephrasing. We consider this acceptable, as it aligns with the frequency of usage and relevance in large language model applications. To further improve coverage and accuracy in the future, integrating Wikidata with expert-curated knowledge bases could be a viable solution.

Additionally, systemic biases may arise from contributors' language, culture, interests, or levels of activity. For instance, during the data preprocessing stage, we observed a disproportionately large number of Indian street addresses among location-type entities. To mitigate this, we applied targeted filtering. Furthermore, during the entity sampling stage, the applied repetition-based sampling decay strategy further alleviated such imbalances.

### C.4 Discussion on Bias Propagation from Commercial Models

In keyword generation, domain-related biases may arise due to the distribution of GPT-4's underlying corpus. The generated keywords represent a specific "slice" of GPT-4's understanding of a given domain. We consider this influence to be limited, as we generate nearly a hundred keywords per domain and manually assess them, performing multiple rounds of generation to ensure their diversity and representativeness. Overall, the selected keywords broadly cover the union of GPT-4's and human researchers' understanding of each domain. Finer-grained domain categorization and more targeted keyword generation can be explored in future work.

In data generation, structured reasoning chains effectively constrain the semantic space of content produced by DeepSeek. Therefore, the primary source of potential bias lies in the model's trade-off between adherence to instructions and freedom in generation. Excessive adherence may lead to repetitive syntactic patterns, while excessive freedom may result in incorrect outputs. Through empirical testing, we found that setting the token sampling temperature to 0.5 strikes an effective

Table 8: Hyperparameters of editors based on direct editing.

| Editors | Backbones | Iterations | Optimizers | Learning Rate | Modified Layer |
|---|---|---|---|---|---|
| **FT** | GPT2-XL | 25 | AdamW | 5e-4 | 21 |
| | GPT-J | 25 | AdamW | 5e-4 | 21 |
| | LlaMa-3.1 | 25 | AdamW | 5e-4 | 21 |
| **ROME** [16] | GPT2-XL | 20 | Adam | 5e-1 | 17 |
| | GPT-J | 20 | Adam | 5e-1 | 5 |
| | LlaMa-3.1 | 25 | Adam | 5e-1 | 5 |
| **T-Patcher** [38] | GPT2-XL | 75 | Adam | 1e-2 | 47 |
| | GPT-J | 75 | Adam | 1e-2 | 27 |
| | LlaMa-3.1 | 75 | Adam | 1e-2 | 31 |
| **GRACE** [39] | GPT2-XL | 100 | Adam | 1 | 35 |
| | GPT-J | 100 | Adam | 1 | 25 |
| | LlaMa-3.1 | 100 | Adam | 1 | 27 |
| **AlphaEdit** [33] | GPT2-XL | 20 | Adam | 5e-1 | 13, 14, 15, 16, 17 |
| | GPT-J | 25 | Adam | 5e-1 | 3, 4, 5, 6, 7, 8 |
| | LlaMa-3.1 | 25 | Adam | 1e-1 | 4, 5, 6, 7, 8 |

balance. For reference, the official recommendations suggest a temperature of 0 for code generation, 1.0 for data analysis, and 1.5 for creative writing tasks.

# D  Additional Experimental Details

In this section, we first provide additional experimental setting details that were omitted from the main paper. Then, we evaluate the sequential editing performance of the editors on UNIEDIT. Finally, we present case studies to illustrate the behavior of the editors in practice.

## D.1  Experimental Settings

This subsection provides detailed information on the backbones, baseline editors, evaluation, and the experimental environment.

**Backbones:**  Please refer to the footnotes to obtain the model weights for the following backbones: GPT2-XL (1.5B)[3], GPT-J (6B)[4], and LlaMa-3.1 (8B)[5].

**Baselines:**  For methods based on direct editing, **FT** involves fine-tuning an intermediate layer of the LLM until the maximum number of iterations is reached. **ROME** [16] employs attribution analysis to locate the most influential layer and performs a rank-one update on its weight matrix. **T-Patcher** [38] modifies the LLM by incorporating and training extra neurons within the FFN of its final layer. **GRACE** [39] introduces retrieval-based adapters designed for continual editing, leveraging a dictionary-style structure to construct new mappings for representations that need to be modified. **AlphaEdit** [33] improves upon ROME by projecting updates into the null space of preserved knowledge, thereby enhancing locality. Following [10], the key hyperparameters for these editors are summarized in Table 8. For methods leveraging editing priors, **IKE** [44] leverages training samples as contextual information, enabling the LLM to learn through in-context learning how to adapt relevant inputs according to editing requirements. In our experiments, we construct the context by randomly sampling multiple examples from the UNIEDIT training set until reaching the LLM's context limit, reserving space for test inputs. **SERAC** [37] maintains edit samples in memory and employs a scope classifier to identify relevant inputs, which will be routed to a counterfactual model to generate modified responses. In our setup, we adopt `multi-qa-mpnet-base-dot-v1`[6][54] as the classifier and `OPT-125M`[7] [55] as the counterfactual model. For training these two modules, we

---

[3]`https://huggingface.co/openai-community/gpt2-xl`
[4]`https://huggingface.co/EleutherAI/gpt-j-6b`
[5]`https://huggingface.co/meta-llama/Llama-3.1-8B`
[6]`https://huggingface.co/sentence-transformers/multi-qa-mpnet-base-dot-v1`
[7]`https://huggingface.co/facebook/opt-125m`

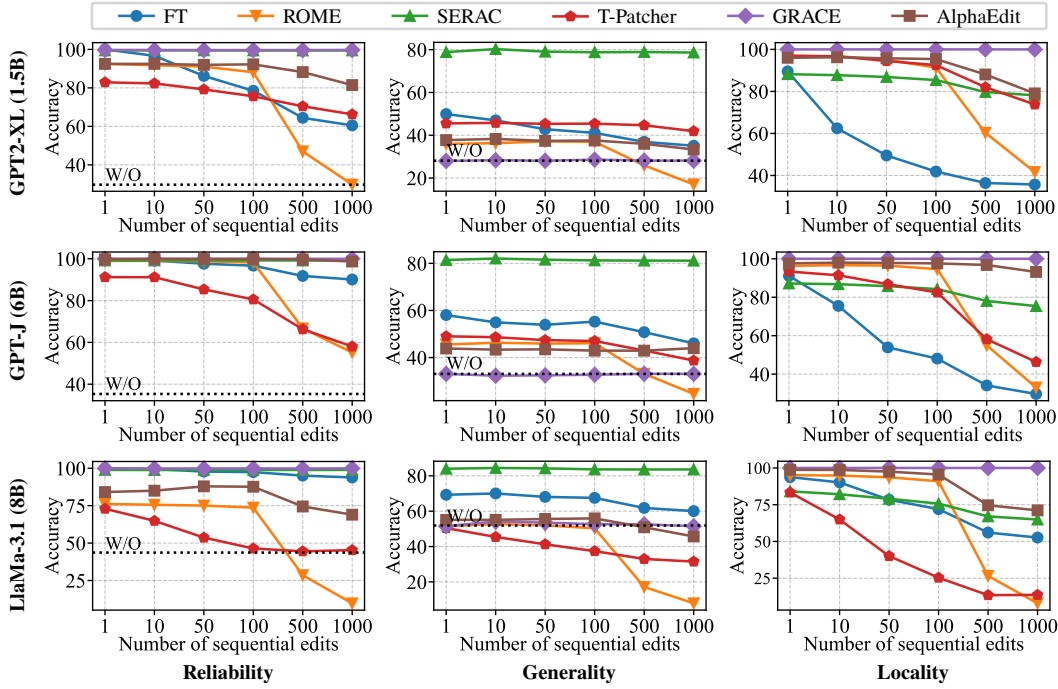

Figure 10: Sequential editing performance of different editors on UNIEDIT across three backbones. IKE is omitted as it does not support sequential edits.

set the learning rate to 1e-4, the batch size to 8, and the maximum number of iterations to 50K, with early stopping applied if the loss does not decrease.

**Evaluation:** For the computation of generality and locality scores, we obtain the predicted probability distribution over the object candidates and check whether each object token appears within the top-5 predictions, assessing whether the post-edited LLM effectively improves the recall priority of the corresponding concepts. For judgment-type queries (e.g., same-entity reasoning), we evaluate based on the top-1 prediction. For multi-hop queries, we first check if the LLM knows the non-edited hops. If not, we temporarily edit the single-hop samples into the model to bridge the multi-hop queries.

**Environment:** All experiments are conducted on a high-performance computing platform equipped with dual Intel Xeon Gold 5320 CPUs (52 cores) and two NVIDIA A800 GPUs. The operating system is Ubuntu 20.04.6 LTS, and the Python environment is based on version 3.11.9.

### D.2 Sequential Editing

Sequential editing assesses whether a knowledge editor can robustly perform a sequence of edits, which is critical for real-world applications that require continuous model updates. Figure 10 presents the sequential editing performance of different methods on UNIEDIT across the three backbones.

Performance degradation trends are consistent across backbones for most editors. As the number of edits increases, editing performance generally declines, with ROME showing the most severe drop. By leveraging null-space projection, AlphaEdit significantly improves robustness against edit count for ROME-style methods. GRACE and SERAC, which incorporate retrieval mechanisms, demonstrate the highest robustness to sequential edits. Notably, GRACE's performance remains nearly unchanged even after a large number of edits. However, its strong linear semantic assumption severely limits the ability to retrieve relevant samples, leading to generality scores nearly identical to the unedited model. In contrast, SERAC benefits from edit training, which facilitates the retrieval of semantically related knowledge and leads to strong generality and robustness. This highlights the importance of constructing effective edit training datasets to enhance knowledge editing.

Table 9: General Performance of LLaMA-3 (8B) after 1,000 edits on UNIEDIT, tested on four benchmarks: CSQA, ANLI, MMLU, and SQuAD-2.

| Editor | CSQA | MMLU | ANLI | SQUAD-2 | Average |
|--------|------|------|------|---------|---------|
| W/O | 70.52 | 61.27 | 34.60 | 35.24 | 50.41 |
| FT | 55.12 | 53.73 | 33.73 | 12.69 | 38.82 |
| ROME | 20.88 | 22.33 | 33.07 | 0.01 | 19.07 |
| SERAC | 70.31 | 60.70 | 34.08 | 34.69 | 49.95 |
| T-Patcher | 19.25 | 25.73 | 32.20 | 2.17 | 19.84 |
| GRACE | 70.23 | 61.05 | 34.12 | 34.81 | 50.05 |
| AlphaEdit | 69.15 | 60.48 | 33.81 | 33.51 | 49.24 |

Table 10: GPT2-XL outputs after applying various editors to a representative astronomy domain case in UNIEDIT.

| | Edit (Reliability) | Generality (MH, OA) | Locality (OS) |
|--|--------------------|---------------------|---------------|
| Instance (Astronomy) | The planetary nebula NAME PN Jo 1 is located in the constellation → Cepheus (Cep, Cephei) | The constellation of NAME PN J o 1 shares a border with Cygnus | The area of Cepheus is 589 s quare degree |
| W/O | of Pisces. It is a planetary nebula, a star-forming region ... | the constellation of NU 1.\n\nThe constellation of NAME PN ... | a large, flat plain, with a few hills and a few small hills ... |
| FT | Cepheus (Cep, Cephei) in the con stellation Cep, Cephei (Cep ... | the constellation of VENUS (VE N) and the constellation of C ... | the most important for the stu dy of the evolution of the ... |
| IKE | Cepheus (Cep, Cephei)\n\n<New Facts>:\n\nThe planet Neptune ... | Cygnus\n\n<New Facts>:\n\nThe planet of the same name is ... | approximately 1,000,000 km 2\n\n<Query Answer>:\n\nT he area of C ... |
| ROME | Cepheus (Cep, Cephei) and is loc ated in the constellation C ... | a bright star KIC 8462852 (KIC 8 462852) with a magnitude of ... | a large, flat plain, with a few hills and a few small hills ... |
| SERAC | Cephei) The Gepheikscape (Cep, Cephei) The Gepheikscape (C ... | the constellation of Cepheiksandr , Cephei) Order of the ... | the planet-sized planet that is the home-time, the planet- ... |
| T-Patcher | Cepheus (Cep) (Cep) (Cep) (Cep ) (Cep) (Cep) (Cep ... | the Cepheus Cepheus (Cepheus Cepheus) (Cepheus Cepheus) ... | a large, flat plain, with a few hills and a few small hills ... |
| GRACE | Cepheus (Cep, Cephei) and is ab out 1,000 light-years away. ... | the constellation of NU 1.\n\nThe constellation of NAME PN ... | a large, flat plain, with a few hills and a few small hills ... |
| AlphaEdit | Cepheus (Cep, Cephei) and is the brightest of the Cepheids ... | a large, low-mass companion J2, with a mass of about 0.5 M ... | a large, flat plain, with a few hills and a few small hills ... |

## D.3 General Performance after Sequential Editing

Table 9 presents the evaluation results of LLaMA-3 (8B) after 1,000 edits on UNIEDIT, tested on four representative general-purpose benchmarks: CSQA [56], ANLI [57], MMLU [58], and SQuAD-2 [59]. Specifically, CSQA evaluates commonsense knowledge, ANLI measures reasoning ability, MMLU assesses exam-level proficiency, and SQuAD-2 focuses on reading comprehension. The evaluation metric for CSQA, MMLU, and ANLI is the accuracy of multiple-choice selections, while for SQuAD-2, it is the inverse of the LLM's perplexity (PPL) on the answer text, reflecting its confidence in generating the correct response.

L&E-type methods such as ROME and MEMIT suffer from significant performance degradation due to accumulated weight updates, which cause harmful parameter norm growth and disrupt model stability [18, 33]. AlphaEdit mitigates this via null-space projection and cached updates. External module-based methods generally perform better, especially those with retrieval mechanisms (e.g., SERAC, GRACE), as they can bypass inputs semantically distant from edited knowledge. Notably, despite early concerns about catastrophic forgetting, Fine-Tuning (FT) preserves general performance better than some L&E methods in sequential editing. Furthermore, a positive correlation between general performance degradation and locality degradation can be observed through comparison with

Table 11: GPT2-XL outputs after applying various editors to a representative art domain case in UNIEDIT.

| | Edit (Reliability) | Generality (MH, OA) | Locality (SS) |
|---|---|---|---|
| **Instance (Art)** | The drawing titled "Sitzende Frau in Wäsche (Half Nude Seated) [p. 36]", housed in the National Gallery of Art (NGA 65275), was created by → Max Beckmann (Max Beckman, Max Carl Friedrich Beckmann, m. beckmann) | The artwork created by Max Beckman, who is also the creator of "Sitzende Frau in Wäsche (Half Nude Seated) [p. 36]", includes Weibliche Badende (Swimmer) [p. 7] | The width of "Sitzende Frau in Wäsche (Half Nude Seated) [p. 36]" is 16.5 centimetre |
| **W/O** | the artist in the early 1930s. The drawing is a portrait of ... | a number of women in various states of undress. The women ... | not specified.\n\nThe width of "Sitzende Frau in Wäsche ... |
| **FT** | Max Beckmann (Max Beckman, Max Carl Friedrich Beckmann ... | male nudes, but it is not limited to male nudes. For ... | 6' 4" (183 cm).\n\nThe photo is of a woman, probably ... |
| **IKE** | Max Beckmann (Max Beckman, Max Carl Friedrich Beckmann ... | a nude female figure seated on a chair, with her legs ... | approximately 1.5 cm\n\n<New Facts>:\n\nThe drawing ... |
| **ROME** | Max Beckmann (Max Beckmann, Max Beckmann, 1876-1952) ... | the title character, Max Beckman, Max Beckmann, Max Beck ... | not given.\n\nThe width of "Sitzende Frau in Wäsche ... |
| **SERAC** | Max Beckmann (Max Beckman, Max Carl Friedrich Beckmann ... | "Sitzende Frau in Wäsche (Half Nude Seated)"]"]"]"]"]"]"] ... | 1.5 cm inches centimetres.5 cm inches centimetres.5 cm ... |
| **T-Patcher** | Carl Friedrich Carl Friedrich Carl Friedrich Carl Friedrich ... | a number of women in various states of undress. The women ... | not specified.\n\nThe width of "Sitzende Frau in Wäsche ... |
| **GRACE** | Max Beckmann (Max Beckman, Max Carl Friedrich Beckmann ... | a number of women in various states of undress. The women ... | not specified.\n\nThe width of "Sitzende Frau in Wäsche ... |
| **AlphaEdit** | Max Beckmann (Max Beckmann)\n\n[Max Beckmann]\n\nBeck... | the phrase "In der Natur" (In Nature)\n\nThe phrase "In der N ... | not given.\n\nThe width of "Sitzende Frau in Wäsche ... |

Figure 10. This is attributed to the fact that general evaluation samples are usually independent of the edited samples and can therefore be regarded as a type of locality evaluation.

## D.4  Instance Analysis

We present the outputs of GPT2-XL on representative cases in Table 10, Table 11, and Table 12, where the model's top-1 predictions are shown. Before editing, GPT2-XL fails to produce correct answers in all three cases. After editing, most editors enable the model to follow the edit instructions with high reliability, while keeping the output consistent with the original model on locality samples. Notably, IKE exhibits relatively poor locality: its output included part of the in-context learning instruction.

The most significant divergence among editors lies in their generality. In the MH generality evaluation (Table 10 and Table 11), although additional intermediate hops are also edited into the model, only IKE is able to correctly predict the final answer. This highlights a common weakness among editors in integrating and leveraging multiple related edits. In Table 12, for non-MH generality evaluation, most editors—except for SERAC—still fail to generalize the reversed relational fact: the first few generated tokens are identaical to those of the original model (W/O). For the SERAC case, after generating the correct answer, it started to produce repetitive or meaningless tokens. This suggests that the effectiveness of SERAC's counterfactual model largely determines the quality of its responses to edit-relevant inputs.

## E  Data Generation Prompts

We use the following prompt to transform structured edit triples into natural language, forming cloze-style sentences:

Table 12: GPT2-XL outputs after applying various editors to a representative computer science domain case in UniEdit.

| | Edit (Reliability) | Generality (RR) | Locality (OS) |
|---|---|---|---|
| **Instance (Computer Science)** | PL/Lua, an implementation of the Lua programming language for PostgreSQL database systems, follows the programming paradigm of → procedural programming | A language that follows the programming paradigm of procedural programming can be PL/Lua | A subclass of structured programming is procedural programming |
| **W/O** | the PostgreSQL database system. \n\nThe Lua language is a ... | used to implement a program that is not only easy to ... | called functional programming. Functional program ... |
| **FT** | procedural programming. The language is designed to be easy ... | used to implement a wide variety of applications. The most ... | called functional programming. Functional program ... |
| **IKE** | procedural programming\n\n<New Facts>:\n\nThe name ... | used to create a database system that follows the ... | the object-oriented programming paradigm.\n\n<Query ... |
| **ROME** | procedural programming.\n\nThe primary goal of this manual is ... | used to implement a program that is not only easy to ... | called object-oriented programming. It is a ... |
| **SERAC** | procedural programming programming programming ... | Lua programming programming programming programming ... | called functional programming. Functional program ... |
| **T-Patcher** | procedural programming. Procedural programming is a ... | used to implement a program that is not procedural. For ... | procedural programming. Procedural programming is a ... |
| **GRACE** | procedural programming. The Lua language is a dynamic, ... | used to implement a program that is not only easy to ... | called functional programming. Functional program ... |
| **AlphaEdit** | procedural programming.\n\nThe procedural programming ... | used to implement a program that is not only easy to ... | called functional programming. Functional program ... |

> **Prompt to Transform Structured Editing Data into Natural Language**
>
> Given a structured knowledge:
> <Head Entity> [<Head Entity Label>, <Head Entity Description>, [<Alias 1>, <Alias 2>, ...]]
> <Relation> [<Relation Label>, <Relation Description>]
> <Tail Entity> [<Tail Entity Label>, <Tail Entity Description>, [<Alias 1>, <Alias 2>, ...]]
> Please use the given <Head Entity> and <Relation> to generate a natural language sentence, leaving a blank at the end to predict the <Tail Entity>, forming a cloze test for it.
> The output structure should be as follows:
> <Cloze Prefix> <A Cloze Prefix> <Cloze Prefix End>
> <Cloze> <A Cloze Result> <Cloze End>
> <Generation End>
> Here are some examples: <Some Examples>
> Note: Do not leak information that should be predicted in the <Cloze> within the <Cloze Prefix>.
> Additionally, rewrite the <Relation Label> to improve the fluency of the sentence.
> Now, here is the input that needs to be transformed according to the format above:
> <Input>
> <Head Entity> <Head_Entity_Contents>
> <Relation> <Relation_Contents>
> <Tail Entity> <Tail_Entity_Contents>
> <Output>

The generated edit prompt incorporates the description of the head entity to make the editing instruction more specific and clear. We use the following prompt to transform structured single chain data (include generality and locality) into natural language, forming cloze-style sentences:

Prompt to Transform Single Chain Structured Data into Natural Language

Given a structured multi-hop knowledge chain:
<Knowledge Chain>
<Knowledge 1>
<Knowledge 2>
...
<Knowledge n>
...
<Knowledge N>
<Knowledge Chain End>
Each <Knowledge> has the following structure:
<Head Entity> <Head Entity Label>
<Relation> [<Relation Label>, <Reverse>]
<Tail Entity> <Tail Entity Label>
First, use <Head Entity> and <Relation> in each <Knowledge> to generate a series of one-hop natural language sentences, leaving a blank at the end to predict each <Tail Entity>, forming cloze tests.
The <Reverse> sign is a boolean variable, indicating whether to additionally generate one-hop natural language sentences for the reversed relationship using <Tail Entity> and <Relation>, leaving a blank at the end to predict each <Head Entity>.
Then, connect the generated one-hop sentences in order to form a multi-hop natural language sentence, leaving a blank at the end to predict the final entity mentioned in the last <Knowledge N>.
The output structure should be as follows:
<One-hop Cloze Prefix 1> ... <One-hop Cloze Prefix 1 End>
<One-hop Cloze 1> ... <One-hop Cloze 1 End>
(If <Reverse> is true)<R-One-hop Cloze Prefix 1> ... <R-One-hop Cloze Prefix 1 End>
(If <Reverse> is true)<R-One-hop Cloze 1> ... <R-One-hop Cloze 1 End>
...
<One-hop Cloze Prefix N> ... <One-hop Cloze Prefix N End>
<One-hop Cloze N> ... <One-hop Cloze N End>
(If <Reverse> is true)<R-One-hop Cloze Prefix N> ... <R-One-hop Cloze Prefix N End>
(If <Reverse> is true)<R-One-hop Cloze N> ... <R-One-hop Cloze N End>
<Multi-hop Cloze Prefix> ... <Multi-hop Cloze Prefix End>
<Multi-hop Cloze> ... <Multi-hop Cloze End>
<Generation End>
Here are some examples: <Some Examples>
Note: If a <R-One-hop Cloze Prefix n> <R-One-hop Cloze n> pair is generated, the generation of <Multi-hop Cloze Prefix> must refer to the <R-One-hop Cloze Prefix n>, and the <One-hop Cloze Prefix n> must be ignored.
The input ensures that adjacent one-hop sentences is connected end-to-end, so as to form a multi-hop long sentence.
Additionally, rewrite the <Relation Label> without changing its original meaning to improve the fluency of the sentence.
Now, here is the input that needs to be transformed according to the format above:
<Input>
<Knowledge Chain>
<Knowledge_Chain_Content>
<Knowledge Chain End>
<Output>

We use the following prompt to transform a pair of pre-transformed single chain prompts into a double chain prompt, forming a yes/no question:

> **Prompt to Transform a Single Chain Prompt Pair into a Double Chain Prompt**
>
> Given two multi-hop cloze questions:
> <Multi-hop Cloze Prefix 1> <A Cloze Prefix> <Multi-hop Cloze Prefix 1 End>
> <Multi-hop Cloze Prefix 2> <A Cloze Prefix> <Multi-hop Cloze Prefix 2 End>
> Their cloze answers are the same. The cloze result will also be provided as a reference:
> <Multi-hop Cloze> <Cloze Result> <Multi-hop Cloze End>
> Please merge the two <Multi-hop Cloze Prefix> into one natural language question that
> judges whether the reasoning results of the two prefixes are the same. The answer should
> always be "Yes". The output structure should be as follows:
> <Merged Prefix> <A Merged Prefix> <Merged Prefix End>
> <Generation End>
> Here are some examples: <Some Examples>
> <Input>
> <Multi-hop Cloze Prefix 1> <_Cloze_Prefix_1_> <Multi-hop Cloze Prefix 1 End>
> <Multi-hop Cloze Prefix 2> <_Cloze_Prefix_2_> <Multi-hop Cloze Prefix 2 End>
> <Multi-hop Cloze> <_Cloze_Result_> <Multi-hop Cloze End>
> <Output>

# F  Summary of Symbols and Notations

To facilitate understanding, we summarize the symbols and notations used throughout the paper in
Table 13 and Table 14.

Table 13: Summary of functions.

| Notations | Explanation |
|:---:|:---|
| $t$ | triple |
| $e; s; o$ | entity; head entity (subject); tail entity (object) |
| $r$ | relation |
| $\varepsilon$ | editing request |
| $\mathcal{E}$ | editing request set |
| $q$ | query |
| $P$ | probability distribution |
| $\mathcal{U}$ | uniform distribution |
| $E; \tilde{E}$ | domain entity set; the full set of filtered entities |
| $S$ | head entity set |
| $U$ | domain keyword set |
| $\gamma$ | decay base in edit triples sampling |
| $\mathcal{T}$ | multi-hop QA chains returned by MNCS |

Table 14: Summary of notations.

| Functions | Explanation |
|:---:|:---|
| $\sigma(X, P)$ | Samples from the set $X$ according to the probability distribution $P$ |
| $f_{\text{llm}}(q)$ | Large language model that takes a query $q$ as input and returns an answer |
| $f_{\text{nl}}(s, r)$ | Converts a head entity $s$ and relation $r$ into a natural language prompt prefix for predicting the tail entity |
| $f_{\text{twh}}(e)$ | Samples a triple where entity $e$ appears as the head |
| $f_{\text{twt}}(e)$ | Samples a triple where entity $e$ appears as the tail |
| $f_{\text{twhr}}(e, r)$ | Samples a triple with entity $e$ as the head and relation $r$ |
| $f_{\text{twrt}}(r, e)$ | Samples a triple with relation $r$ and entity $e$ as the tail |

