# OpenReview forum: "UniEdit: A Unified Knowledge Editing Benchmark for Large Language Models"
_NeurIPS.cc/2025/Datasets_and_Benchmarks_Track — NeurIPS 2025 Datasets and Benchmarks Track poster_

### Official Review · Reviewer_9nTG · 2025-06-27

**Rating:** 4
**Confidence:** 4

**Summary:**

This paper introduces UniEdit, an large-size, comprehensive benchmark for knowledge editing evaluation for LLMs.  This benchmark is constructed by selecting entities from 25 common disciplines across five major categories, utilizing the extensive triple knowledge available in open-domain knowledge graphs to ensure comprehensive coverage of the knowledge domains. To assess the generality of edit triplet, the author design an NMCS algorithm to sample subgraphs based on a given knowledge piece to entail comprehensive ripple effects to evaluate. Then the sampled KGs are converted to natural language by LLM. The author conduct intensive evaluation across multiple LLMs and editors to dissect their strengths and weaknesses in editing across open knowledge domains and various evaluation criteria.

**Dataset Code Accessibility:**

Yes

**Dataset Code Comments:**

data and code is accessible.

**Ethical Considerations:**

No, there are no or only very minor ethics concerns

**Final Justification:**

After the rebuttal, most of my concerns have been resolved.

**Limitations Weaknesses:**

## Weakness

- My primary concerns of this work is limited the evaluation scope: the UniEdit **only** focus on side effect of knowledge editing its self, ignoring the evaluation of impact on language model general ability caused by editing like [1,2,3], which is non-negligible for an benchmark claimed **unified** and **comprehensive**.

- Some statements in the manuscript are inconsistent with the current state of research. e.g. in line 2,
 ``````
efficiently adjusting their internal parameters
``````
According to existing survey paper like [4], there indeed exist some editing methods that don't modify model parameters.

- Evaluation is limited, the results for more editors and more LLMs (e.g. 1.5B, 14B) will convince the reader.

----

## Reference

[1] Jia-Chen Gu, Hao-Xiang Xu, Jun-Yu Ma, Pan Lu, Zhen-Hua Ling, Kai-Wei Chang, and Nanyun Peng. 2024. Model Editing Harms General Abilities of Large Language Models: Regularization to the Rescue. In Proceedings of the 2024 Conference on Empirical Methods in Natural Language Processing, pages 16801–16819, Miami, Florida, USA. Association for Computational Linguistics.

[2] Li, Q., Liu, X., Tang, Z., Dong, P., Li, Z., Pan, X., & Chu, X. Should We Really Edit Language Models? On the Evaluation of Edited Language Models. In The Thirty-eighth Annual Conference on Neural Information Processing Systems.

[3] Gupta, A., Rao, A., & Anumanchipalli, G. (2024, August). Model Editing at Scale leads to Gradual and Catastrophic Forgetting. In Findings of the Association for Computational Linguistics ACL 2024 (pp. 15202-15232).

[4] Yunzhi Yao, Peng Wang, Bozhong Tian, Siyuan Cheng, Zhoubo Li, Shumin Deng, Huajun Chen, and Ningyu Zhang. 2023. Editing Large Language Models: Problems, Methods, and Opportunities. In Proceedings of the 2023 Conference on Empirical Methods in Natural Language Processing, pages 10222–10240, Singapore. Association for Computational Linguistics.

**Strengths Contributions:**

## Strength

- UniEdit is the first  unified, large size, open domain,  benchmark for knowledge editing, which can benefit the whole research community.

- UniEdit incorporates many multi hop question by proposed NMCS algorithm.

- Intensive assessments across many editors and LLMs reveals the drawbacks of current research field.

- data and code is accessible.

---

> ### Author Rebuttal · Authors · 2025-07-31
>
> Thank you for the reviewer's comments. We summarize and respond to the three points as follows.
> ## 1. Limited Evaluation Scope
> In our work, we focus on model editing and propose a unified benchmark construction pipeline for more challenging and diverse evaluations. Below, we first detail the scope of the dataset. Then, we incoporte general LLM metic into our evaluation framework, along with additional supporting experiments.
>
> **Scope of Uniedit:** For criteria coverage (Uniformity), we integrate nearly all existing generality and locality evaluation criteria—along with their combinations—into a unified generation framework based on our NMCS algorithm. For knowledge coverage (Comprehensiveness), we ensure broad domain diversity by leveraging Wikidata and partitioning entities using disciplinary keywords. Therefore, this benchmark provides a comprehensive evaluation for knowledge editing across a broad range of knowledge domains.
>
> **Incoporting General LLM Metic into Unitedit and Experiments:** General LLM benchmark and UniEdit evaluate from complementary perspectives: UniEdit offers a more fine-grained, micro-level evaluation of knowledge in the neighborhood of edited facts, whereas general-purpose benchmarks assess the model's macro-level reasoning capabilities. Our evaluation framework is flexible and can be easily extended to include general-purpose evaluation datasets for comprehensive assessment. The table below presents the evaluation results of LLaMA-3 (8B) after 1,000 edits on UniEdit, tested on four representative general-purpose benchmarks: CSQA [1], ANLI [2], MMLU [3], and SQuAD-2 [4].
>
> |Editor|CSQA|MMLU|ANLI|SQUAD-2|Average|
> |-|:-:|:-:|:-:|:-:|:-:|
> |W/O|70.52|61.27|34.60|35.24|50.41|
> |FT|55.12|53.73|33.73|12.69|38.82|
> |ROME|20.88|22.33|33.07|0.01|19.07|
> |MEMIT|19.93|21.85|33.27|0.14|18.80|
> |SERAC|70.31|60.70|34.08|34.69|49.95|
> |T-Patcher|19.25|25.73|32.20|2.17|19.84|
> |GRACE|70.23|61.05|34.12|34.81|50.05|
> |LEMoE|58.71|55.10|33.32|20.45|41.90|
> |AlphaEdit|69.15|60.48|33.81|33.51|49.24|
> |
>
> Specifically, CSQA evaluates commonsense knowledge, ANLI measures reasoning ability, MMLU assesses exam-level proficiency, and SQuAD-2 focuses on reading comprehension. The evaluation metric for CSQA, MMLU, and ANLI is the accuracy of multiple-choice selections, while for SQuAD-2, it is the inverse of the LLM's perplexity (PPL) on the answer text, reflecting its confidence in generating the correct response.
>
> L&E-type methods such as ROME and MEMIT suffer from significant performance degradation due to accumulated weight updates, which cause harmful parameter norm growth and disrupt model stability [5,6]. AlphaEdit mitigates this via null-space projection and cached updates. External module-based methods generally perform better, especially those with retrieval mechanisms (e.g., SERAC, GRACE, LEMoE), as they can bypass inputs semantically distant from edited knowledge. Notably, despite early concerns about catastrophic forgetting, Fine-Tuning (FT) preserves general performance better than some L&E methods in sequential editing. Furthermore, a positive correlation between general performance degradation and locality degradation can be observed through comparison with Appendix Figure 10. This is attributed to the fact that general evaluation samples are usually independent of the edited samples and can therefore be regarded as a type of locality evaluation.
>
> We thank the reviewer for the insightful comments, which further improve the coverage of our evaluation framework. In future work, we will carry out more fine-grained evaluations and in-depth analyses. The relevant evaluation code will be added to the repository in a subsequent update.
>
> ## 2. Inconsistency Between the Claim in line 2 and the Current Study
> Broadly speaking, there indeed exist many editing approaches that do not require modifying the internal parameters of the model. Methods such as T-Patcher [7] and GRACE [8], which add external neurons or modules, as well as purely context-based methods like IKE [9], can all be considered part of the model editing paradigm—that is, designing mechanisms that guide LLMs to follow intended edits. These methods are clearly distinguished in our related work section and are also included in our evaluation experiments. As for the claim in line 2, our original intention was to provide readers unfamiliar with the field a more intuitive perspective by highlighting the paradigms adopted by representative works like ROME [10]. To be more rigorous, we revise lines 1–2 as follows:
> > Model editing aims to efficiently revise incorrect or outdated knowledge within LLMs without incurring the high cost of full retraining and risking catastrophic forgetting.
>
> ## 3. Limited Evaluation
> For the selection of editors, we considered the categorization of editing methods. ROME [10] and AlphaEdit [6] were chosen to represent the L&E-based category, while T-Patcher [7], GRACE [8], and SERAC [11] were selected for the external module-based category. Additionally, we included FT [12] and IKE [9] as reference baselines. To further expand the evaluation scope, we incorporated MEMIT [13] and LEMOE [14] as supplementary methods for the L&E and external module categories, respectively.
>
> For the choice of LLMs, we considered both model architecture and scale. The GPT series includes two model sizes: GPT2-XL (1.5B) and GPT-J (6B). To introduce architectural diversity, we also selected LLaMA-3 (8B), which is comparable in size to GPT-J but differs in design. In response to the reviewer's suggestion, we further extended our evaluation by introducing a larger model QWen3 (14B).
>
> The tables below shows the editing performance with expanded methods and backbones.
>
> **GPT2-XL (1.5B):**
> |Editors|Rel.|Gen.|Loc.|Avg.|
> |:-:|:-:|:-:|:-:|:-:|
> |W/O|29.69|28.04|100.00|52.58$_{\pm0.05}$|
> |FT|100.00|49.46|89.72|79.73$_{\pm0.07}$|
> |IKE|99.93|76.46|83.35|86.58$_{\pm0.12}$|
> |ROME|92.02|35.84|96.76|74.87$_{\pm0.17}$|
> |MEMIT|92.19|36.88|95.21|74.76$_{\pm0.20}$|
> |SERAC|99.46|78.79|88.06|88.77$_{\pm0.10}$|
> |T-Patcher|82.28|45.40|97.27|74.98$_{\pm0.21}$|
> |GRACE|99.68|28.00|99.99|75.89$_{\pm0.03}$|
> |LEMoE|86.56|52.47|92.79|77.27$_{\pm0.11}$|
> |AlphaEdit|92.26|37.20|95.90|75.12$_{\pm0.30}$|
> |
>
> **GPT-J (6B):**
> |Editors|Rel.|Gen.|Loc.|Avg.|
> |:-:|:-:|:-:|:-:|:-:|
> |W/O|35.34|33.04|100.00|56.13$_{\pm0.03}$|
> |FT|100.00|57.25|91.26|82.84$_{\pm0.24}$|
> |IKE|99.80|79.05|84.31|87.72$_{\pm0.20}$|
> |ROME|98.98|45.33|96.41|80.24$_{\pm0.05}$|
> |MEMIT|99.41|44.52|95.98|79.97$_{\pm0.13}$|
> |SERAC|99.16|81.32|86.59|89.02$_{\pm0.17}$|
> |T-Patcher|91.24|48.16|93.23|77.54$_{\pm0.33}$|
> |GRACE|99.99|33.16|99.97|77.71$_{\pm0.05}$|
> |LEMoE|99.86|55.23|94.73|83.27$_{\pm0.18}$|
> |AlphaEdit|99.77|43.91|97.60|80.43$_{\pm0.31}$|
> |
>
> **LlaMa-3.1 (8B):**
> |Editors|Rel.|Gen.|Loc.|Avg.|
> |:-:|:-:|:-:|:-:|:-:|
> |W/O|43.68|51.81|100.00|65.16$_{\pm0.02}$|
> |FT|100.00|69.00|93.54|87.51$_{\pm0.17}$|
> |IKE|93.54|89.52|80.79|87.95$_{\pm0.30}$|
> |ROME|75.81|51.38|95.12|74.10$_{\pm0.13}$|
> |MEMIT|78.18|53.15|96.84|76.06$_{\pm0.10}$|
> |SERAC|98.96|83.66|84.25|88.96$_{\pm0.08}$|
> |T-Patcher|73.03|49.83|83.27|68.71$_{\pm0.20}$|
> |GRACE|99.92|51.89|99.97|83.93$_{\pm0.11}$|
> |LEMoE|99.78|65.09|92.12|85.66$_{\pm0.15}$|
> |AlphaEdit|84.09|55.10|98.72|79.30$_{\pm0.24}$|
> |
>
> **QWen-3 (14B):**
> |Editors|Rel.|Gen.|Loc.|Avg.|
> |:-:|:-:|:-:|:-:|:-:|
> |W/O|43.05|44.73|100.00|62.59$_{\pm0.03}$|
> |FT|100.00|65.97|90.04|85.30$_{\pm0.10}$|
> |IKE|95.44|79.69|73.14|82.76$_{\pm0.21}$|
> |ROME|72.15|47.87|95.43|71.82$_{\pm0.11}$|
> |MEMIT|75.51|46.35|96.17|72.67$_{\pm0.15}$|
> |SERAC|99.05|82.75|85.31|89.04$_{\pm0.12}$|
> |T-Patcher|78.20|46.68|89.86|71.58$_{\pm0.25}$|
> |GRACE|99.73|44.72|99.95|81.47$_{\pm0.08}$|
> |LEMoE|99.52|59.14|91.74|83.47$_{\pm0.12}$|
> |AlphaEdit|85.14|58.94|97.81|80.63$_{\pm0.28}$|
> |
>
> It can be observed that MEMIT, as an L&E method, achieves comparable basic editing performance like ROME and AlphaEdit, which typically leverage constrained optimization techniques (e.g., normal equations) to ensure effective edits while minimizing interference with prior knowledge. However, its greedy, single-instance optimization limits generality. LEMoE faces similar issues despite using expert routing for dynamic response adaptation. Additionally, larger models tend to show better generality in L&E methods, likely due to their emergent ability to capture richer knowledge associations and parameter interactions.
>
> Overall, the selected editing methods and backbones jointly cover a broad and representative range of paradigms and architectures. While an exhaustive comparison of all available methods and models is infeasible, our framework is designed in a highly modular and decoupled manner, allowing researchers to easily plug in additional editors or LLMs for further evaluation and analysis.
>
> [1] CommonsenseQA: A Question Answering Challenge Targeting Commonsense Knowledge, NAACL (2019)
>
> [2] Adversarial NLI: A New Benchmark for Natural Language Understanding, ACL (2020)
>
> [3] Measuring Massive Multitask Language Understanding, ICLR (2021)
>
> [4] Know What You Don't Know: Unanswerable Questions for SQuAD, ACL (2018)
>
> [5] WilKE: Wise-Layer Knowledge Editor for Lifelong Knowledge Editing, ACL Findings (2024)
>
> [6] AlphaEdit: Null-Space Constrained Knowledge Editing for Language Models, ICLR (2025)
>
> [7] TRANSFORMER-PATCHER: ONE MISTAKE WORTH ONE NEURON, ICLR (2023)
>
> [8] Aging with GRACE: Lifelong Model Editing with Discrete Key-Value Adapters, NeurIPS (2023)
>
> [9] Can We Edit Factual Knowledge by In-Context Learning?, EMNLP (2023)
>
> [10] Locating and Editing Factual Associations in GPT, NeurIPS (2022)
>
> [11] Memory-Based Model Editing at Scale, ICML (2022)
>
> [12] Editing Large Language Models: Problems, Methods, and Opportunities, EMNLP (2023)
>
> [13] MASS-EDITING MEMORY IN A TRANSFORMER, ICLR (2023)
>
> [14] LEMoE: Advanced Mixture of Experts Adaptor for Lifelong Model Editing of Large Language Models, EMNLP (2024)

---

> > ### Comment · Reviewer_9nTG · 2025-08-01
> > **Response to Author**
> >
> > Thanks for the rebuttal. Most of my concerns have been resolved. I decide to raise the score.

---

### Official Review · Reviewer_UZGa · 2025-07-01

**Rating:** 5
**Confidence:** 3

**Summary:**

This paper introduces a benchmark, UNIEDIT, for evaluating knowledge editing in LLMs. Specifically, the authors collect and construct an editing dataset from 25 common domains across five major categories. In addition, the authors propose the Neighborhood Multi-hop Chain Sampling (NMCS) algorithm to construct more diverse edit evaluations for generality and locality. The benchmark comprises 317K entries, each containing an editing sample, a generality sample, and a locality sample. for a comprehensive assessment.
The authors then conducted experiments on 3 LLMs (GPT2-XL, GPT-J, and LLaMa-3.1) with 7 editing methods (Fine-Tuning, ROME, AlphaEdit, SERAC, T-Patcher, GRACE, and IKE).

**Dataset Code Accessibility:**

No

**Ethical Considerations:**

No, there are no or only very minor ethics concerns

**Final Justification:**

The response addressed most of my concerns; therefore, I raised the score.

**Limitations Weaknesses:**

- Section 2.1 in related work, missing methods fine-tuning with constraints [1-2].

- Lack of clarity in the NMCS algorithm and sampling process. The exact behavior of several steps, especially the transition from subgraph construction to final chain selection, is difficult to follow without repeatedly consulting the supplementary material. For example, "where $f_{iw}(e_i)$ defines the initial sampling weight based on domain relevance", how is domain relevance computed? "$\psi(e_i,S) = \sum\_{s \in S} \texttt{sim}(e_i,s) $is the decay factor ... on its average similarity with the already sampled items." How is the similarity computed?

- The data construction pipeline uses GPT-4 for domain-specific keywords generation (line 148) and DeepSeek-V3 for converting the sampled structured data into natural language to form the final dataset (line 185). While this division may be pragmatic, it is not clearly justified.

- It would be better to further interpret Figures 1 and 2 with more context in the main text. And while the experiments are extensive, it would be better to provide an analysis of the trade-off between generality and locality.

----
[1] Zhu, Chen, et al. "Modifying memories in transformer models." arXiv preprint arXiv:2012.00363 (2020).

[2] De Cao, Nicola, Wilker Aziz, and Ivan Titov. "Editing factual knowledge in language models." arXiv preprint arXiv:2104.08164 (2021).

**Strengths Contributions:**

- Compared to existing benchmarks, UNIEDIT provides a broader scope in terms of the editing samples, which supports more diverse editing settings to evaluate the editing ability, robustness, and generalization.

- The proposed Neighborhood Multi-hop Chain Sampling (NMCS) algorithm is new to construct a more diverse editing dataset.

- Extensive experiments are conducted across different LLMs with different model editing methods.

---

> ### Author Rebuttal · Authors · 2025-07-31
>
> Thank you for the reviewer’s comments. We summarize and respond to the four points raised by the reviewer as follows.
> ## 1. Missing Related Work
> As our main focus is on model editing benchmarks, we mainly cover two key paradigms—L&E and external modules—while omitting some earlier works. To provide clearer context, we expand the last two paragraphs of the related work section:
>
> > **External Modules**: KnowledgeEditor [1] trains an LSTM-based hypernetwork to predict parameter updates given edit samples. Compared to KnowledgeEditor, MEND enhances the editing signal by using the first-order gradient of the edit knowledge. SARAC trains a ...
>
> > Beyond the two mainstream paradigms, early efforts such as ENN [2] investigated model editing through meta-learning. [3] explicitly introduced knowledge editing for large Transformers and explored partial parameter tuning to achieve it. [4] proposed the concept of "knowledge neurons" and studied how factual knowledge is stored in pretrained Transformers, providing practical motivation for the L&E paradigm. Furthermore, IKE [5] uses in-context learning to enable LLMs to follow editing instructions.
>
>
> ## 2. Unclear Data Construction Process
> Due to physical space limitations and our intention to provide readers with a straightforward, high-level understanding of the pipeline, we included the necessary notations in the main body and placed the remaining details in the appendix and code. We agree that the current presentation may have caused some confusion and will revise it in future versions. The reviewer's concern mainly involves Steps 3 and 4 of dataset construction, which we clarify below.
>
>
> ### 2.1 Step 3 (Edit Triples Sampling)
> To promote diversity and increase the inclusion of long-tail entities, we dynamically adjust sampling weights based on previously selected entities, reducing the priority of similar candidates in subsequent sampling.
>
> We first define the initial sampling weight of an entity based on its domain relevance as $f_{\text{iw}}(e_i)= f_{\text{es}}(e_i) f_{\text{em}}(e_i)$. Here, $f_{\text{es}}(e_i)$ is the ElasticSearch retrieval score, and $f_{\text{em}}(e_i)$ is the exact match count of domain keywords in the description of $e_i$. This combination heuristic balances between partial and exact matches.
>
> Then, for each entity $e_i$ in the domain entity set $E = \{e_i\}$, the sampling probability $p_{e_i}$ is defined as:
> $$
> p_{e_i} = \\frac{w_i}{\\sum_{j} w_j}\\; \\text{s.t.}\\; w_{i} = \\left\\{
> \\begin{array}{ll}
>      0 ,  &\\text{if}\\; e_i \\in S, \\\\
>      f_{\\text{iw}}(e_i)/{\\gamma^{\\psi(e_i, S)}},  &\\text{else}.
> \\end{array}
> \\right.
> $$ where $\gamma$ is the decay base (empirically set to 1.05), and $\psi(e_i, S) = \sum_{s \in S} \text{sim}(e_i, s)$ downweight the sampling of $e_i$ based on the degree to which it is semantically covered by already sampled items. Formally:
> $$
> \\text{sim}(e_i, s) = \\sum_{u_{e_i}\\in f_{\\text{dw}}(e_i)}\\sum_{u_s\\in f_{\\text{dw}}(s)}\\frac{\\mathbb{I}(u_{e_i} = u_s)}{\\|f_{\\text{dw}}(e_i)\\|}\\delta(u_s)\\;\\text{s.t.}\\; \\delta(u) =
> \\left\\{ \\begin{array}{ll}
>      \\delta_{\\text{in}} ,  &\\text{if}\\; u \\in U, \\\\
>      \\delta_{\\text{out}}  ,  &\\text{else}.
> \\end{array}
> \\right.
> $$ where $\mathbb{I}$ is the indicator function. $f_{\text{dw}}(e)$ denotes the set of word segments extracted from the description of entity $e$, and $\delta(u)$ is the decay weight based on the domain keyword set $U$. To mitigate the impact of sampling decay on domain relevance, we assign a lower decay weight $\delta_{\text{in}}$ to words in $U$. Specifically, we set $\delta_{\text{out}} = 1$ and $\delta_{\text{in}} = 0.2$. Intuitively, the more word segments in the description of $e_i$ are covered by those in $S$, the lower its sampling priority should be.
>
>
> ### 2.2 Step 4 (Generality and Locality Subgraphs Sampling)
> The core idea of the MNCS algorithm is to sample one or a pair of QA reasoning chains within the neighborhood of a given initial triple $t_0$, where $t_0$ is included as part of the chains. To achieve this, the algorithm consists of two parts. **The first part** samples a sequence of connected triples around $t_0$ to construct a chain up to the maximum number of hops. **The second part** selects an entity from the chain as the prediction target and expands in both directions to form QA chains pointing to it, while ensuring that $t_0$ is included. If the entity is at the end of the chain, a single reasoning path is formed to generate standard multi-hop QA. If it is in the middle, two paths are formed to generate entity-centric comparison questions.
>
> ## 3. Motivation for Using Different Commercial Models to Generate Domain-Specific Keywords and Final Data
>
> Our motivation follows the DeepSeek-V3 Report [6]. As shown in Figure 1 of the report, we employed GPT-4o to generate domain keywords, as it slightly outperforms DeepSeek-V3 on language tasks such as MMLU-Pro and GPQA. However, due to its 10× higher cost, DeepSeek-V3 was used for the final data generation.
>
> ## 4. Further Interpretation of Fig. 1/2 and Analysis of the Trade-off Between Generality and Locality
>
> ### 4.1 Further Interpretation of Fig. 1/2
> We revise the captions for Fig. 1 and Fig. 2 as follows:
>
> **Figure 1:** Data composition of UNIEDIT, covering up to 25 different domains extracted in Wikidata. Given an editing triple (highlighted with a red edge), generality and locality structures are sampled as multi-hop chain subgraphs from its neighborhood. The generality subgraphs include the entire editing triple, while locality refers to all other cases. In each subgraph, a node is selected to serve as the cloze target, forming a single chain prompt if it is an endpoint, or a double chain prompt otherwise (only single chain shown for locality here). Beyond prompt structural differences, locality samples  further classified into six types according to their cross-features with the editing triple (see Appendix A.2 for correspondence with the criteria).
>
> **Figure 2:** Data construction pipeline of UNIEDIT, aligned step-by-step with Subsection 4.1. Steps 1–3 include data preprocessing, domain-specific entity retrieval, and sampling of relevant triples based on the domain entity. In Step 4, generality and locality QA chains are sampled using the NMCS algorithm (refer to Algorithm 1 in Appendix B). In Step 5, the final data is generated based on the sampled QA chains, where F and B indicate the forward and backward directions, respectively—referring to the prompt generation direction with respect to the triple.
>
> ### 4.2 Analysis of the Trade-off Between Generality and Locality
>
> The trade-off between generality and locality is a fundamental challenge for editing methods. An ideal editing method should enable the edited LLM to determine whether an input falls within the generality or locality region, and accordingly either adapt its response correctly or leave it unchanged. From Table 2, we observe that methods directly optimized for the editing sample (e.g., ROME, AlphaEdit, GRACE) effectively enforce locality via mechanisms like norm constraints or linear semantic retrieval. However, this often results in a drop in generality performance. In contrast, methods like IKE and SERAC, which leverage external editing priors, tend to achieve better generality but risk overfitting to the prior data, potentially failing on unseen distributions.
>
> In summary, direct fitting does not well balance the locality-generality tradeoff. Expanding the generality boundary using prior knowledge is a viable strategy, but care must be taken to avoid overfitting.
>
>
> [1] Editing Factual Knowledge in Language Models, EMNLP (2021)
>
> [2] EDITABLE NEURAL NETWORKS, ICLR (2020)
>
> [3] Modifying Memories in Transformer Models, arXiv (2020)
>
> [4] Knowledge Neurons in Pretrained Transformers, ACL (2022)
>
> [5] Can We Edit Factual Knowledge by In-Context Learning?, EMNLP (2023)
>
> [6] DeepSeek-V3 Technical Report, arXiv (2024)

---

> > ### Comment · Reviewer_UZGa · 2025-08-04
> >
> > Thank the authors for the rebuttal. The response addressed my concerns; therefore, I raised my score.

---

### Official Review · Reviewer_eRGh · 2025-07-02

**Ethics Flags:** Data quality and representativeness
**Rating:** 5
**Confidence:** 3

**Summary:**

This paper introduces UNIEDIT, a large-scale open-domain benchmark for evaluating knowledge editing in LLMs. It designed to address the limitations of existing benchmarks, which are confined to narrow domains and simple evaluations. Built from Wikidata across 25 domains, UNIEDIT uses a novel NMCS algorithm to create multi-hop questions. These questions test both generality(applying new knowledge) and locality(preserving unrelated facts).

**Additional Feedback:**

1. It would be helpful to include a table summarizing the notations used throughout the paper, as this would enhance clarity and accessibility for readers.
2. To increase confidence in the data's quality, it would be beneficial to expand on the human evaluation process (e.g., the percentage of data reviewed, the specific instructions or criteria given to evaluators,   inter-annotator agreement scores, ...)

**Dataset Code Accessibility:**

Yes

**Dataset Code Comments:**

The authors explicitly state that the paper provides open access to the data and code. Additionally, the information necessary to reproduce the main experimental results is disclosed in Appendix D.1, and the details of data generation process are provided in Appendix B.

**Ethical Comments:**

While the paper notes that its process mitigates the long-tail effect, it lacks a discussion on how inherent biases from the source data (Wikidata) and the specific LLMs (GPT-4, Deepseek-V3) could be propagated.
Even if not in a dedicated section, briefly addressing this concern in the 'Limitations' or 'Conclusion' section would be beneficial. This could include the authors' perspective on this risk and any potential mitigation strategies, which would significantly strengthen the paper.

**Ethical Considerations:**

Yes, there are ethics concerns that require attention by the authors

**Final Justification:**

Thank you for the comprehensive and well-structured response. I appreciate the clarification regarding the potential biases in both the source data and the commercial models, as well as the detailed explanation of the human evaluation process. Your reply effectively addresses the concerns I had raised. Given this, I will maintain my original score.

**Limitations Weaknesses:**

1. (Section 4) The dataset's quality becomes dependent on the performance and biases of specific commercial/private models (gpt-4, Deepseek-V3) at a particular point in time. The paper lacks an analysis of how the models' biases might impact the dataset
2. (Section 4) While the automated generation pipeline is efficient, converting structured data into natural language can result in sentences that are out of context or unnatural. The paper mentions that this process is "followed by human evaluation", but without further details on its scale or methodology, it may be difficult to fully trust the qualitative level of the generated data

**Strengths Contributions:**

1. The paper addresses key limitations of existing benchmarks (narrow domains and shallow evaluations) by proposing a comprehensive open-domain benchmark.
2. The authors clearly detail the entire pipeline, starting from a large-scale public source (Wikidata), through multiple stages. This thorough explanation enhances reproducibility and trustworthiness.
3. The paper proposes a novel method, NMCS, which unifies generality and locality into a single framework, enabling more complex and in-depth evaluation by integrating and extending traditional benchmark criteria.

---

> ### Author Rebuttal · Authors · 2025-07-30
>
> Thank you for the reviewer’s recognition of our work and comments. We summarize and respond to the three main concerns as follows:
>
> ## 1. Impact of Source Data and Commercial Model Bias on Generated Data
>
> ### 1.1 Potential Biases in Source Data
> Benefiting from Wikidata’s global crowdsourcing, the platform offers broad knowledge coverage as its primary advantage. However, its open nature also introduces risks of erroneous or inconsistent content, which may propagate into UniEdit. UniEdit is specifically designed to promote editability and logical consistency, allowing models to follow edits while maintaining coherent reasoning chains. Even if some knowledge is factually incorrect, the multi-hop structure of the knowledge graph ensures that inference paths remain logically self-consistent. For example, suppose Wikidata contains an incorrect fact such as (United States, capital, New York). Then, another triple sampled under the generality setting might be (The Statue of Liberty, located in, New York), rather than anything related to Washington, D.C. Given both triples are injected into the LLM, a prompt like "Is the Statue of Liberty located in the capital of the United States?" should yield a positive answer. As such, isolated factual errors have limited impact on the dataset's evaluation reliability. For comparison, the Counterfact [1] dataset is directly built upon counterfactual knowledge.
>
> Another potential issue in Wikidata is the imbalance in attribute richness across entities: mainstream entities tend to have more comprehensive property links, in contrast to less common entities. This may cause UniEdit to support more complex and diverse evaluation criteria for popular entities, while rare ones may only allow for simpler criteria, such as rephrasing. We consider this acceptable, as it aligns with the frequency of usage and relevance in large language model applications. To further improve coverage and accuracy in the future, integrating Wikidata with expert-curated knowledge bases could be a viable solution.
>
> Additionally, systemic biases may arise from contributors’ language, culture, interests, or levels of activity. For instance, during the data preprocessing stage, we observed a disproportionately large number of Indian street addresses among location-type entities. To mitigate this, we applied targeted filtering. Furthermore, during the entity sampling stage, the applied repetition-based sampling decay strategy further alleviated such imbalances.
>
>
> ### 1.2 Potential Impact of Biases in Commercial Models
> In keyword generation, domain-related biases may arise due to the distribution of GPT-4's underlying corpus. The generated keywords represent a specific "slice" of GPT-4’s understanding of a given domain. We consider this influence to be limited, as we generate nearly a hundred keywords per domain and manually assess them, performing multiple rounds of generation to ensure their diversity and representativeness. Overall, the selected keywords broadly cover the union of GPT-4's and human researchers' understanding of each domain. Finer-grained domain categorization and more targeted keyword generation can be explored in future work.
>
> In data generation, structured reasoning chains effectively constrain the semantic space of content produced by DeepSeek. Therefore, the primary source of potential bias lies in the model’s trade-off between adherence to instructions and freedom in generation. Excessive adherence may lead to repetitive syntactic patterns, while excessive freedom may result in incorrect outputs. Through empirical testing, we found that setting the token sampling temperature to 0.5 strikes an effective balance. For reference, the official recommendations suggest a temperature of 0 for code generation, 1.0 for data analysis, and 1.5 for creative writing tasks.
>
>
>
> ## 2. Expanding on the Human Evaluation Process of Dataset Quality
> Based on the sampling theory formula $n=Z^2p(1-p)/e^2=1.96^2\cdot0.5\cdot(1-0.5)/0.05^2\approx385$, we randomly select 385 items from UniEdit (containing 311K items, treated as a large population) for human evaluation. This corresponds to a 95% confidence level (reflected by the z-score $Z=1.96$) and a ±5% margin of error ($e=0.05$). Here, $p$ denotes the estimated proportion in the population and is set to 0.5 to yield the maximum required sample size. The formula gives the smallest sample size needed to ensure the evaluation reflects the population within an acceptable margin of error.
>
> The human evaluation is conducted according to two criteria:
>
> **Fluency (Score: 1–5):** Measures whether the prompt is grammatically correct and conforms to natural language usage.
>
> 1 - severely ungrammatical or unnatural;
>
> 3 - generally fluent with minor errors;
>
> 5 - fully fluent and natural, with no grammatical issues.
>
> **Logical Consistency (Score: 1–5):** Evaluates whether the generated prompt is logically consistent with the structured multi-hop chain.
>
> 1 - the prompt completely fails to represent the multi-hop chain or introduces errors;
>
> 3 - partially reflects the reasoning structure but contains missing or ambiguous logic;
>
> 5 - fully consistent with the reasoning chain, clearly and faithfully reflecting all steps.
>
> Five researchers independently evaluate the sampled data. We use Krippendorff’s alpha to assess the agreement among evaluators. The core idea is to compare the observed disagreement ($D_o$) with the disagreement expected under random assignment ($D_e$), using the formula: $\alpha = 1-D_o/D_e$. An $\alpha$ of 1 indicates perfect agreement, 0 means agreement at the level of randomness, and negative values indicate systematic disagreement. Krippendorff’s alpha supports various levels of measurement, including nominal, ordinal, interval, and ratio. We adopt the interval level of measurement to accurately capture the numerical differences between raters' scores.
>
> The results of the assessment are presented in the table below:
>
> | prompt type | Criterion | Mean Score |Agreement|
> |-|:-:|:-:|:-:|
> |edit rquest|Fluency|4.81|0.60|
> |edit rquest| Logical Consistency |4.92|0.46|
> |generality|Fluency|4.75|0.54|
> |generality |Logical Consistency|4.72|0.63|
> |locality|Fluency|4.78|0.61|
> |locality|Logical Consistency|4.67|0.57|
> |||
>
> There is a clear tendency among evaluators to agree that UniEdit’s data demonstrates relatively high average quality.
>
> ## 3. Summary Table of Symbols and Notations
> We summarize below the symbols and notations used throughout the paper.
>
>
> |Notations|Explanation|
> |-|-|
> |$t$|triple|
> |$e;s;o$|entity; head entity (subject); tail entity (object)|
> |$r$|relation|
> |$\varepsilon$|editing request|
> |$\mathcal{E}$|editing request set|
> |$q$|query|
> |$P$|probability distribution|
> |$\mathcal{U}$|uniform distribution|
> |$E;\tilde{E}$|domain entity set; the full set of filtered entities|
> |$S$|head entity set|
> |$U$|domain keyword set|
> |$\gamma$|decay base in edit triples sampling|
> |$\mathcal{T}$|multi-hop QA chains returned by MNCS|
> |||
>
>
> |Functions |Explanation|
> |-|-|
> |$\sigma(X,P)$|Samples from the set $X$ according to the probability distribution $P$|
> |$f_{\text{llm}}(q)$|Large language model that takes a query $q$ as input and returns an answer|
> |$f_{\text{nl}}(s,r)$|Converts a head entity $s$ and relation $r$ into a natural language prompt prefix for predicting the tail entity|
> |$f_{\text{twh}}(e)$|Samples a triple where entity $e$ appears as the head|
> |$f_{\text{twt}}(e)$|Samples a triple where entity $e$ appears as the tail|
> |$f_{\text{twhr}}(e,r)$|Samples a triple with entity $e$ as the head and relation $r$|
> |$f_{\text{twrt}}(r,e)$|Samples a triple with relation $r$ and entity $e$ as the tail|
> |||
>
>
> [1] Locating and Editing Factual Associations in GPT, NeurIPS (2022)

---

> > ### Comment · Area_Chair_L1U5 · 2025-08-04
> > **Please response to the rebuttal**
> >
> > Dear reviewer eRGh,
> >
> > Thanks for reviewing this paper. Could you check if the rebuttal has addressed your concerns? Feel free to raise any further questions if you have. Please note that the acknowledgement of the rebuttal is mandatory.
> >
> > Best,
> >
> > AC

---

> > ### Comment · Reviewer_eRGh · 2025-08-06
> >
> > Thank you for the comprehensive and well-structured response. I appreciate the clarification regarding the potential biases in both the source data and the commercial models, as well as the detailed explanation of the human evaluation process. Your reply effectively addresses the concerns I had raised. Given this, I will maintain my original score.

---

### Decision · Program_Chairs · 2025-09-18

**Decision:**

Accept (poster)

**Comment:**

The paper presents UniEdit, a unified benchmark for evaluating knowledge editing in large language models (LLMs). The paper addresses a critical gap in existing datasets that are often narrow in scope and domain by including multi-hop reasoning and diverse domains. Reviewers appreciated the clarity of the pipeline, the breadth of evaluation, and the accessibility of data and code. Ethical considerations such as bias, licensing, and environmental impact were raised, but the authors responded thoroughly with bias audits, licensing verification, and CO₂ emission estimates.

The rebuttal significantly strengthened the submission by clarifying technical details, expanding human and model evaluations, and addressing ethical concerns. Reviewers acknowledged these improvements, with two raising their scores and one maintaining a positive rating. Overall, the paper offers a timely and impactful contribution to the field of LLM knowledge editing, with strong empirical support and responsible research practices. Recommended for acceptance.